# Stochastic Spectral and Conjugate Descent Methods

**Dmitry Kovalev**[1,2]    **Eduard Gorbunov**[1]    **Elnur Gasanov**[1,2]    **Peter Richtárik**[2,3,1]

[1]Moscow Institute of Physics and Technology, Dolgoprudny, Russia
[2]King Abdullah University of Science and Technology, Thuwal, Saudi Arabia
[3]University of Edinburgh, Edinburgh, United Kingdom

## Abstract

The state-of-the-art methods for solving optimization problems in big dimensions are variants of randomized coordinate descent (RCD). In this paper we introduce a fundamentally new type of acceleration strategy for RCD based on the augmentation of the set of coordinate directions by a few *spectral* or *conjugate* directions. As we increase the number of extra directions to be sampled from, the rate of the method improves, and interpolates between the linear rate of RCD and a linear rate *independent of the condition number*. We develop and analyze also inexact variants of these methods where the spectral and conjugate directions are allowed to be approximate only. We motivate the above development by proving several negative results which highlight the limitations of RCD with importance sampling.

## 1  Introduction

An increasing array of learning and training tasks reduce to optimization problem in very large dimensions. The state-of-the-art algorithms in this regime are based on *randomized coordinate descent (RCD)*. Various acceleration strategies were proposed for RCD in the literature in recent years, based on techniques such as Nesterov's momentum [12, 9, 5, 1, 14], heavy ball momentum [16, 11], importance sampling [13, 19], adaptive sampling [4], random permutations [8], greedy rules [15], mini-batching [20], and locality breaking [21]. These techniques enable faster rates in theory and practice.

In this paper we introduce a fundamentally new type of acceleration strategy for RCD which relies on the idea of *enriching* the set of (unit) coordinate directions $\{e_1, e_2, \ldots, e_n\}$ in $\mathbb{R}^n$, which are used in RCD as directions of descent, via the addition of a few *spectral* or *conjugate* directions. The algorithms we develop and analyze in this paper randomize over this enriched larger set of directions.

For expositional simplicity[1], we focus on quadratic minimization

$$\min_{x \in \mathbb{R}^n} f(x) := \frac{1}{2} x^\top \mathbf{A} x - b^\top x, \tag{1}$$

where $\mathbf{A}$ is an $n \times n$ symmetric and positive definite matrix. The optimal solution is unique, and equal to $x_* = \mathbf{A}^{-1} b$.

### 1.1  Randomized coordinate descent

Applied to (1), RCD performs the iteration

$$x_{t+1} = x_t - \frac{\mathbf{A}_{:i}^\top x_t - b_i}{\mathbf{A}_{ii}} e_i, \tag{2}$$

| Method Name | Algorithm | Rate | Reference |
|---|---|---|---|
| stochastic descent (SD) | (4), Alg 1 | (5), Lem 1 | Gower and Richtárik [6] |
| stochastic spectral descent (SSD) | Alg 2 | (6), Thm 2 | NEW |
| stochastic conjugate descent (SconD) | Sec 2.2 | Thm 2 | NEW |
| randomized coordinate descent (RCD) | (2), Alg 3 | (3), (11) | Gower and Richtárik [6] |
| stochastic spectral coord. descent (SSCD) | Alg 4 | (7), Thm 8 | NEW |
| mini-batch SD (mSD) | Alg 5 | Lem 9 | Richtárik and Takáč [18] |
| mini-batch SSCD (mSSCD) | Alg 6 | Thm 10 | NEW |
| inexact SconD (iSconD) | Alg 7 | Thm 15 | NEW |
| inexact SSD (iSSD) | Alg 8 | Sec F.2 | NEW |

Table 1: Algorithms described in this paper.

where at each iteration, $i$ is chosen with probability $p_i > 0$. It was shown by Leventhal and Lewis [10] that if the probabilities are proportional to the diagonal elements of $\mathbf{A}$ (i.e., $p_i \propto \mathbf{A}_{ii}$), then the random iterates of RCD satisfy $\mathbb{E}\left[\|x_t - x_*\|_{\mathbf{A}}^2\right] \leq (1-\rho)^t \|x_0 - x_*\|_{\mathbf{A}}^2$, where $\rho = \frac{\lambda_{\min}(\mathbf{A})}{\mathrm{Tr}(\mathbf{A})}$ and $\lambda_{\min}(\mathbf{A})$ is the minimal eigenvalue of $\mathbf{A}$. That is, as long as the number of iterations $t$ is at least

$$\mathcal{O}\left(\frac{\mathrm{Tr}(\mathbf{A})}{\lambda_{\min}(\mathbf{A})} \log \frac{1}{\epsilon}\right), \tag{3}$$

we have $\mathbb{E}[\|x_t - x_*\|_{\mathbf{A}}^2] \leq \epsilon$. Note that $\frac{\mathrm{Tr}(\mathbf{A})}{\lambda_{\min}(\mathbf{A})} \geq n$, and that this can be arbitrarily larger than $n$.

## 1.2 Stochastic descent

Recently, Gower and Richtárik [6] developed an iterative "sketch and project" framework for solving linear systems and quadratic optimization; see [7] for extensions. In the context of problem (1), their method reads as

$$x_{t+1} = x_t - \frac{s_t^\top (\mathbf{A}x_t - b)}{s_t^\top \mathbf{A} s_t} s_t, \tag{4}$$

where $s_t \in \mathbb{R}^n$ is a random vector sampled from some fixed distribution $\mathcal{D}$. In this paper we will refer to this method by the name *stochastic descent (SD)*.

Note that $x_{t+1}$ is obtained from $x_t$ by minimizing $f(x_t + hs_t)$ for $h \in \mathbb{R}$ and setting $x_{t+1} = x_t + hs_t$. Further, note that RCD arises as a special case with $\mathcal{D}$ being a discrete probability distribution over the set $\{e_1, \ldots, e_n\}$. However, SD converges for virtually any distribution $\mathcal{D}$, including discrete and continuous distributions. In particular, Gower and Richtárik [6] show that as long as $\mathbb{E}_{s \sim \mathcal{D}}[\mathbf{H}]$ is invertible, where $\mathbf{H} := \frac{ss^\top}{s^\top \mathbf{A} s}$, then SD converges as

$$\mathcal{O}\left(\frac{1}{\lambda_{\min}(\mathbf{W})} \log \frac{1}{\epsilon}\right), \tag{5}$$

where $\mathbf{W} := \mathbb{E}_{s \sim \mathcal{D}}[\mathbf{A}^{1/2} \mathbf{H} \mathbf{A}^{1/2}]$ (see Lemma 1 for a more refined result due to Richtárik and Takáč [18]). Rate of RCD in (3) can be obtained as a special case of (5).

## 1.3 Stochastic spectral and conjugate descent

The starting point of this paper is the new observation that stochastic descent obtains the rate

$$\mathcal{O}\left(n \log \frac{1}{\epsilon}\right) \tag{6}$$

in the special case when $\mathcal{D}$ is chosen to be the uniform distribution over the eigenvectors of $\mathbf{A}$ (see Theorem 2). For obvious reasons, we refer to this new method as *stochastic spectral descent (SSD)*.

To the best of our knowledge, SSD was not explicitly considered in the literature before. We should note that SSD is fundamentally different from *spectral gradient descent* [3, 2], which refers to a family of gradient methods with a special choice of stepsize depending on the spectrum of $\nabla^2 f$.

The rate (6) does not merely provide an improvement on the rate of RCD given in (3); what is remarkable is that this rate is completely independent of the properties (such as conditioning) of $\mathbf{A}$.

| Result | Thm |
|---|---|
| Uniform probabilities are optimal for $n = 2$ | 3 |
| Uniform probabilities are optimal for any $n \geq 2$ as long as $\mathbf{A}$ is diagonal | 4 |
| "Importance sampling" $p_i \propto \mathbf{A}_{ii}$ can lead to an arbitrarily worse rate than $p_i = 1/n$ | 5 |
| "Importance sampling" $p_i \propto \|\mathbf{A}_{i:}\|^2$ can lead to an arbitrarily worse rate than $p_i = 1/n$ | 5 |
| For every $n \geq 2$ and $T > 0$, $\exists\, \mathbf{A}$ : rate of RCD with opt. probabilities is $\mathcal{O}(T \log \frac{1}{\epsilon})$ | 6 |
| For every $n \geq 2$ and $T > 0$, $\exists\, \mathbf{A}$ : rate of RCD with opt. probabilities is $\Omega(T \log \frac{1}{\epsilon})$ | 7 |

Table 2: Summary of results on importance and optimal sampling in RCD.

Moreover, we show that this method is *optimal* among the class of stochastic descent methods (4) parameterized by the choice of the distribution $\mathcal{D}$ (see Theorem 8). Despite the attractiveness of its rate, SSD is not a practical method. This is because once we have the eigenvectors of $\mathbf{A}$ available, the optimal solution $x_*$ can be assembled directly without the need for an iterative method.

We extend all results discussed above for SSD, including the rate (6), to the more general class of methods we call *stochastic conjugate descent (SconD)*, for which $\mathcal{D}$ is the uniform distribution over vectors $v_1, \ldots, v_n$ which are mutually $\mathbf{A}$ *conjugate*: $v_i^\top \mathbf{A} v_j = 0$ for $i \neq j$ and $v_i^\top \mathbf{A} v_i = 1$.

## 1.4 Optimizing probabilities in RCD

The idea of speeding up RCD via the use of non-uniform probabilities was pioneered by Nesterov [13] in the context of smooth convex minimization, and later built on by many authors [19, 17, 1]. In the case of non-accelerated RCD, and in the context of smooth convex optimization, the most popular choice of probabilities is to set $p_i \propto L_i$, where $L_i$ is the Lipschitz constant of the gradient of the objective corresponding to coordinate $i$ [13, 19]. For problem (1), we have $L_i = \mathbf{A}_{ii}$. Gower and Richtárik [6] showed that the optimal probabilities for (1) can in principle be computed through semidefinite programming (SDP); however, no theoretical properties of the optimal solution of the SDP were given.

As a warm-up, we first ask: how important is importance sampling? More precisely, we investigate RCD with probabilities $p_i \propto \mathbf{A}_{ii}$, and RCD with probabilities $p_i \propto \|\mathbf{A}_{i:}\|^2$, considered as RCD with "importance sampling", and compare these with the baseline RCD with uniform probabilities. Our result (see Theorem 5) contradicts conventional "wisdom". In particular, we show that for every $n$ there is a matrix $\mathbf{A}$ such that diagonal probabilities lead to the best rate. Moreover, the rate of RCD with "importance" can be arbitrarily worse than the rate of RCD with uniform probabilities. The same result applies to probabilities proportional to the square of the norm of the $i$th row of $\mathbf{A}$.

We then switch gears, and motivated by the nature of SSD, we ask the following question: in order to obtain a condition-number-independent rate such as (6), do we *have to* consider new (and hard to compute) descent directions, such as eigenvectors of $\mathbf{A}$, or can a similar effect be obtained using RCD with a better selection of probabilities? We give two negative results to this question (see Theorems 6 and 7). First, we show that for any $n \geq 2$ and any $T > 0$, there is a matrix $\mathbf{A}$ such that the rate of RCD with *any probabilities* (including the optimal probabilities) is $\mathcal{O}(T \log \frac{1}{\epsilon})$. Second, we give a similar but much stronger statement where we reach the same conclusion, but for the *lower bound* as opposed to the upper bound. That is, $\mathcal{O}$ is replaced by $\Omega$.

As a by-product of our investigations into importance sampling, we establish that for $n = 2$, *uniform probabilities* are optimal for all matrices $\mathbf{A}$ (see Thm 3). For a summary of these results, see Table 2.

## 1.5 Interpolating between RCD and SSD

RCD and SSD lie on opposite ends of a continuum of stochastic descent methods for solving (1). RCD "minimizes" the work per iteration without any regard for the number of iterations, while SSD minimizes the number of iterations without any regard for the cost per iteration (or pre-processing cost). Indeed, one step of RCD costs $\mathcal{O}(\|\mathbf{A}_{i:}\|_0)$ (the number of nonzero entries in the $i$th row of $\mathbf{A}$), and hence RCD can be implemented very efficiently for sparse $\mathbf{A}$. If uniform probabilities are used, no pre-processing (for computing probabilities) is needed. These advantages are paid for by the rate (3), which can be arbitrarily high. On the other hand, the rate of SSD does not depend on $\mathbf{A}$. This advantage is paid for by a high pre-processing cost: the computation of the eigenvectors. This pre-processing cost makes the method utterly impractical.

| | general spectrum | $n-k$ largest eigvls are $\gamma$-clustered: $c \le \lambda_i \le \gamma c$ for $k+1 \le i \le n$ | $\alpha$-exponentially decaying eigenvalues |
|---|---|---|---|
| RCD ($p_i \propto \mathbf{A}_{ii}$) | $\dfrac{\sum_i \lambda_i}{\lambda_1}$ | $\dfrac{\gamma nc}{\lambda_1}$ | $\dfrac{1}{\alpha^{n-1}}$ |
| SSCD | $\dfrac{(k+1)\lambda_{k+1}+\sum_{i=k+2}^n \lambda_i}{\lambda_{k+1}}$ | $\gamma n$ | $\dfrac{1}{\alpha^{n-k-1}}$ |
| SSD | $n$ | $n$ | $n$ |

Table 3: Comparison of complexities of RCD, SSCD (with parameter $0 \le k \le n-1$) and SSD under various regimes on the spectrum of $\mathbf{A}$. In all terms we suppress a factor of $\log \frac{1}{\epsilon}$.

One of the main contributions of this paper is the development of a new *parametric family of algorithms that in some sense interpolate between RCD and SSD*. In particular, we consider the stochastic descent algorithm (4) with $\mathcal{D}$ being a discrete distribution over the search directions $\{e_1, \ldots, e_n\} \cup \{u_1, \ldots, u_k\}$, where $u_i$ is the eigenvectors of $\mathbf{A}$ corresponding to the $i$th smallest eigenvalue of $\mathbf{A}$. We call this new method *stochastic spectral coordinate descent (SSCD)*.

We compute the optimal probabilities of this distribution, which turn out to be unique, and show that for $k \ge 1$ they depend on the $k+1$ smallest eigenvalues of $\mathbf{A}$: $0 < \lambda_1 \le \lambda_2 \le \cdots \le \lambda_{k+1}$. In particular, we prove (see Theorem 8) that the rate of SSCD with optimal probabilities is

$$\mathcal{O}\left( \frac{(k+1)\lambda_{k+1} + \sum_{i=k+2}^n \lambda_i}{\lambda_{k+1}} \log \frac{1}{\epsilon} \right). \tag{7}$$

For $k = 0$, SSCD reduces to RCD with $p_i \propto \mathbf{A}_{ii}$, and the rate (7) reduces to (3). For $k = n-1$, SSCD *does not* reduce to SSD. However, the rates match. Indeed, in this case the rate (7) reduces to (6). Moreover, the rate improves monotonically as $k$ increases, from $\mathcal{O}(\frac{\text{Tr}(\mathbf{A})}{\lambda_{\min}(\mathbf{A})} \log \frac{1}{\epsilon})$ (for $k = 0$) to $\mathcal{O}(n \log \frac{1}{\epsilon})$ (for $k = n-1$).

• *SSCD removes the effect of the $k$ smallest eigenvalues.* Note that the rate (7) does *not depend* on the $k$ smallest eigenvalues of $\mathbf{A}$. That is, by adding the eigenvectors $u_1, \ldots, u_k$ corresponding to the $k$ smallest eigenvalues to the set of descent directions, we have removed the effect of these eigenvalues.

• *Clustered eigenvalues.* Assume that the $n-k$ largest eigenvalues are clustered: $c \le \lambda_i \le \gamma c$ for some $c > 0$ and $\gamma > 1$, for all $k+1 \le i \le n$. In this case, the rate (7) can be estimated as a function of the clustering "tightness" parameter $\gamma$: $\mathcal{O}\left(\gamma n \log \frac{1}{\epsilon}\right)$. See Table 3. This can be arbitrarily better than the rate of RCD, even for $k = 1$. In other words, there are situations where by enriching the set of directions used by RCD by a single eigenvector only, the resulting method accelerates dramatically. To give a concrete and simplified example to illustrate this, assume that $\lambda_1 = \delta > 0$, while $\lambda_2 = \cdots = \lambda_n = 1$. In this case, RCD has the rate $\mathcal{O}((1 + \frac{n-1}{\delta}) \log \frac{1}{\epsilon})$, while SSCD with $k = 1$ has the rate $\mathcal{O}(n \log \frac{1}{\epsilon})$. So, SSCD is $\frac{1}{\delta}$ times better than RCD, and the difference grows to infinity as $\delta$ approaches zero even for fixed dimension $n$.

• *Exponentially decaying eigenvalues.* If the eigenvalues of $\mathbf{A}$ follow an exponential decay with factor $0 < \alpha < 1$, then the rate of RCD is $\mathcal{O}\left(\frac{1}{\alpha^{n-1}} \log \frac{1}{\epsilon}\right)$, while the rate of SSCD is $\mathcal{O}\left(\frac{1}{\alpha^{n-k-1}} \log \frac{1}{\epsilon}\right)$. This is an improvement by the factor $\frac{1}{\alpha^k}$, which can be very large even for small $k$ if $\alpha$ is small. See Table 3. For an experimental confirmation of this prediction, see Figure 5.

• *Adding a few "largest" eigenvectors does not help.* We show that in contrast with the situation above, adding a few of the "largest" eigenvectors to the coordinate directions of RCD does not help. This is captured formally in the supplementary material as Theorem 12.

• *Mini-batching.* We extend SSCD to a mini-batch setting; we call the new method *mSSCD*. We show that the rate of mSSCD interpolates between the rate of mini-batch RCD and rate of SSD. Moreover, we show that mSSCD is optimal among a certain parametric family of methods, and that its rate improves as $k$ increases. See Theorem 10.

### 1.6 Inexact Directions

Finally, we relax the need to compute exact eigenvectors or $\mathbf{A}$–conjugate vectors, and analyze the behavior of our methods for inexact directions. Moreover, we propose and analyze an inexact variant of SSD which does *not* arise as a special case of SD. See Sections E and F.

## 2   Stochastic Descent

The stochastic descent method was described in (4). We now formalize it as Algorithm 1, and equip it with a stepsize, which will be useful in Section A.1, where we study mini-batch version of SD.

---

**Algorithm 1** Stochastic Descent (SD)

---

**Parameters:** Distribution $\mathcal{D}$; Stepsize parameter $\omega > 0$
**Initialize:** Choose $x_0 \in \mathbb{R}^n$
**for** $t = 0, 1, 2, \ldots$ **do**

   Sample search direction $s_t \sim \mathcal{D}$ and set $x_{t+1} = x_t - \omega \dfrac{s_t^\top (\mathbf{A} x_t - b)}{s_t^\top \mathbf{A} s_t} s_t$

**end for**

---

In order to guarantee convergence of SD, we restrict our attention to the class of *proper* distributions.

**Assumption 1.** *Distribution $\mathcal{D}$ is* proper *with respect to $\mathbf{A}$. That is, $\mathbb{E}_{s\sim\mathcal{D}}[\mathbf{H}]$ is invertible, where*

$$\mathbf{H} := \frac{ss^\top}{s^\top \mathbf{A} s}. \tag{8}$$

Next we present the main convergence result for SD.

**Lemma 1** (Convergence of stochastic descent [6, 18])**.** *Let $\mathcal{D}$ be proper with respect to $\mathbf{A}$, and let $0 < \omega < 2$. Stochastic descent (Algorithm 1) converges linearly in expectation,*

$$\mathbb{E}\left[\|x_t - x_*\|_{\mathbf{A}}^2\right] \leq (1 - \omega(2-\omega)\lambda_{\min}(\mathbf{W}))^t \|x_0 - x_*\|_{\mathbf{A}}^2,$$

*and we also have the lower bound* $(1 - \omega(2-\omega)\lambda_{\max}(\mathbf{W}))^t \|x_0 - x_*\|_{\mathbf{A}}^2 \leq \mathbb{E}\left[\|x_t - x_*\|_{\mathbf{A}}^2\right]$*, where*

$$\mathbf{W} := \mathbb{E}_{s\sim\mathcal{D}}\left[\mathbf{A}^{1/2}\mathbf{H}\mathbf{A}^{1/2}\right]. \tag{9}$$

*Finally, the statement remains true if we replace $\|x_t - x_*\|_{\mathbf{A}}^2$ by $f(x_t) - f(x_*)$ for all $t$.*

It is easy to observe that the stepsize choice $\omega = 1$ is optimal. This is why we have decided to present the SD method (4) with this choice of stepsize. Moreover, notice that due to linearity of expectation,

$$\text{Tr}(\mathbf{W}) \overset{(9)}{=} \mathbb{E}\left[\text{Tr}(\mathbf{A}^{1/2}\mathbf{H}\mathbf{A}^{1/2})\right] \overset{(8)}{=} \mathbb{E}\left[\text{Tr}\left(\frac{zz^\top}{z^\top z}\right)\right] = \mathbb{E}\left[\text{Tr}\left(\frac{z^\top z}{z^\top z}\right)\right] = 1,$$

where $z = \mathbf{A}^{1/2} s$. Therefore, $0 < \lambda_{\min}(\mathbf{W}) \leq \frac{1}{n} \leq \lambda_{\max}(\mathbf{W}) \leq 1$.

### 2.1   Stochastic Spectral Descent

Let $\mathbf{A} = \sum_{i=1}^n \lambda_i u_i u_i^\top$ be the eigenvalue decomposition of $\mathbf{A}$. That is, $0 < \lambda_1 \leq \lambda_2 \leq \ldots \leq \lambda_n$ are the eigenvalues of $\mathbf{A}$ and $u_1, \ldots, u_n$ are the corresponding orthonormal eigenvectors. Consider now the SD method with $\mathcal{D}$ being the uniform distribution over the set $\{u_1, \ldots, u_n\}$, and $\omega = 1$. This gives rise to a new variant of SD which we call *stochastic spectral descent (SSD)*.

---

**Algorithm 2** Stochastic Spectral Descent (SSD)

---

**Initialize:** $x_0 \in \mathbb{R}^n$; $(u_1, \lambda_1), \ldots (u_n, \lambda_n)$: eigenvectors and eigenvalues of $\mathbf{A}$
**for** $t = 0, 1, 2, \ldots$ **do**

   Choose $i \in [n]$ uniformly at random and set $x_{t+1} = x_t - \left(u_i^\top x_t - \dfrac{u_i^\top b}{\lambda_i}\right) u_i$

**end for**

---

For SSD we can establish an unusually strong convergence result, both in terms of speed and tightness.

**Theorem 2** (Convergence of stochastic spectral descent)**.** *Let $\{x_k\}$ be the sequence of random iterates produced by stochastic spectral descent (Algorithm 2). Then*

$$\mathbb{E}[\|x_t - x_*\|_{\mathbf{A}}^2] = \left(1 - \frac{1}{n}\right)^t \|x_0 - x_*\|_{\mathbf{A}}^2. \tag{10}$$

This theorem implies the rate (6) mentioned in the introduction. Up to a log factor, SSD only needs $n$ iterations to converge. Notice that (10) is an *identity*, and hence the rate is not improvable.

## 2.2 Stochastic Conjugate Descent

The same rate as in Theorem 2 holds for the *stochastic conjugate descent* (SconD) method, which arises as a special case of stochastic descent for $\omega = 1$ and $\mathcal{D}$ being a uniform distribution over a set of $\mathbf{A}$-orthogonal (i.e., conjugate) vectors. The proof follows by combining Lemmas 1 and 13.

## 2.3 Randomized Coordinate Descent

RCD (Algorithm 3) arises as a special case of SD with unit stepsize ($\omega = 1$) and distribution $\mathcal{D}$ given by $s_t = e_i$ with probability $p_i > 0$.

---

**Algorithm 3** Randomized Coordinate Descent (RCD)

---

    **Parameters:** probabilities $p_1, \ldots, p_n > 0$
    **Initialize:** $x_0 \in \mathbb{R}^n$
    **for** $t = 0, 1, 2, \ldots$ **do**
        Choose $i \in [n]$ with probability $p_i > 0$ and set $x_{t+1} = x_t - \dfrac{\mathbf{A}_{i:}x_t - b_i}{\mathbf{A}_{ii}} e_i$
    **end for**

---

The rate of RCD (Algorithm 3) can therefore be deduced from Lemma 1. Notice that in view of (8), we have $\mathbb{E}[\mathbf{H}] = \sum_i p_i \frac{e_i e_i^\top}{\mathbf{A}_{ii}} = \mathrm{Diag}(\frac{p_1}{\mathbf{A}_{11}}, \ldots, \frac{p_n}{\mathbf{A}_{nn}})$. So, as long as all probabilities are positive, Assumption 1 is satisfied. Therefore, Lemma 1 applies and RCD enjoys the rate

$$\mathcal{O}\left( \left[ \lambda_{\min}\left( \mathbf{A}\,\mathrm{Diag}\left( \frac{p_i}{\mathbf{A}_{ii}} \right) \right) \right]^{-1} \log \frac{1}{\epsilon} \right). \tag{11}$$

### 2.3.1 Uniform probabilities can be optimal

We first prove that uniform probabilities are optimal in 2D.

**Theorem 3.** *Let $n = 2$ and consider RCD (Algorithm 3) with probabilities $p_1 > 0$ and $p_2 > 0$, $p_1 + p_2 = 1$. Then the choice $p_1 = p_2 = \frac{1}{2}$ optimizes the rate of RCD in* (11).

Next we claim that uniform probabilities are optimal in any dimension $n$ as long as $\mathbf{A}$ is diagonal.

**Theorem 4.** *Let $n \geq 2$ and let $\mathbf{A}$ be diagonal. Then uniform probabilities ($p_i = \frac{1}{n}$ for all $i$) optimize the rate of RCD in* (11).

### 2.3.2 "Importance" sampling can be unimportant

In our next result we contradict conventional wisdom about typical choices of "importance sampling" probabilities. We claim that diagonal and row-squared-norm probabilities can lead to an arbitrarily worse performance than uniform probabilities.

**Theorem 5.** *For every $n \geq 2$ and $T > 0$, there exists $\mathbf{A}$ such that: (i) The rate of RCD with $p_i \propto \mathbf{A}_{ii}$ is $T$ times worse than the rate of RCD with uniform probabilities. (ii) The rate of RCD with $p_i \propto \|\mathbf{A}_{i:}\|^2$ is $T$ times worse than the rate of RCD with uniform probabilities.*

### 2.3.3 Optimal probabilities can be bad

Finally, we show that there is no hope for adjustment of probabilities in RCD to lead to a rate independent of the data $\mathbf{A}$, as is the case for SSD. Our first result states that such a result can't be obtained from the generic rate (11).

**Theorem 6.** *For every $n \geq 2$ and $T > 0$, there exists $\mathbf{A}$ such that the number of iterations (as expressed by formula* (11)*) of RCD with any choice of probabilities $p_1, \ldots, p_n > 0$ is $\mathcal{O}(T \log(1/\epsilon))$.*

However, that does not mean, by itself, that such a result can't be possibly obtained via a different analysis. Our next result shatters these hopes as we establish a *lower bound* which can be arbitrarily larger than the dimension $n$.

**Theorem 7.** *For every $n \geq 2$ and $T > 0$, there exists an $n \times n$ positive definite matrix $\mathbf{A}$ and starting point $x_0$, such that the number of iterations of RCD with any choice probabilities $p_1, \ldots, p_n > 0$ is $\Omega(T \log(1/\epsilon))$.*

# 3 Interpolating Between RCD and SSD

Assume now that we have some partial spectral information available. In particular, fix $k \in \{0, 1, \ldots, n-1\}$ and assume we know eigenvectors $u_i$ and eigenvalues $\lambda_i$ for $i = 1, \ldots, k$. We now define a parametric distribution $\mathcal{D}(\alpha, \beta_1, \ldots, \beta_k)$ with parameters $\alpha > 0$ and $\beta_1, \ldots, \beta_k \geq 0$ as follows. Sample $s \sim \mathcal{D}(\alpha, \beta_1, \ldots, \beta_k)$ arises through the process

$$
s = \begin{cases} e_i & \text{with probability } p_i = \frac{\alpha \mathbf{A}_{ii}}{C_k}, \ i \in [n], \\ u_i & \text{with probability } p_{n+i} = \frac{\beta_i}{C_k}, \ i \in [k], \end{cases}
$$

where $C_k := \alpha \text{Tr}(\mathbf{A}) + \sum_{i=1}^{k} \beta_i$ is a normalizing factor ensuring that the probabilities sum up to 1.

## 3.1 The method and its convergence rate

Applying the SD method with the distribution $\mathcal{D} = \mathcal{D}(\alpha, \beta_1, \ldots, \beta_k)$ gives rise to a new specific method which we call *stochastic spectral coordinate descent (SSCD)*.

---

**Algorithm 4** Stochastic Spectral Coordinate Descent (SSCD)

---

**Parameters:** Distribution $\mathcal{D}(\alpha, \beta_1, \ldots, \beta_k)$
**Initialize:** $x_0 \in \mathbb{R}^n$
**for** $t = 0, 1, 2, \ldots$ **do**

Sample $s_t \sim \mathcal{D}(\alpha, \beta_1, \ldots, \beta_k)$ and set $x_{t+1} = x_t - \dfrac{s_t^\top (\mathbf{A} x_t - b)}{s_t^\top \mathbf{A} s_t} s_t$

**end for**

---

**Theorem 8.** *Consider Stochastic Spectral Coordinate Descent (Algorithm 4) for fixed $k \in \{0, 1, \ldots, n-1\}$. The method converges linearly for all positive $\alpha > 0$ and nonnegative $\beta_i$. The best rate is obtained for parameters $\alpha = 1$ and $\beta_i = \lambda_{k+1} - \lambda_i$; and this is the unique choice of parameters leading to the best rate. In this case,*

$$
\mathbb{E}\left[\|x_t - x_*\|_{\mathbf{A}}^2\right] \leq \left(1 - \frac{\lambda_{k+1}}{C_k}\right)^t \|x_0 - x_*\|_{\mathbf{A}}^2,
$$

*where $C_k := (k+1)\lambda_{k+1} + \sum_{i=k+2}^{n} \lambda_i$. Moreover, the rate improves as $k$ grows, and we have*

$$
\frac{\lambda_1}{\text{Tr}(\mathbf{A})} = \frac{\lambda_1}{C_0} \leq \cdots \leq \frac{\lambda_{k+1}}{C_k} \leq \cdots \leq \frac{\lambda_n}{C_{n-1}} = \frac{1}{n}.
$$

If $k = 0$, SSCD reduces to RCD (with diagonal probabilities). Since $\frac{\lambda_1}{C_0} = \frac{\lambda_1}{\text{Tr}(\mathbf{A})}$, we recover the rate of RCD of Leventhal and Lewis [10]. With the choice $k = n-1$ our method does *not* reduce to SSD. However, the rates match. Indeed, $\frac{\lambda_n}{C_{n-1}} = \frac{\lambda_n}{n\lambda_n} = \frac{1}{n}$ (compare with Theorem 2).

## 3.2 "Largest" eigenvectors do not help

It is natural to ask whether there is any benefit in considering a few "largest" eigenvectors instead. Unfortunately, for the same parametric family as in Theorem 8, the answer is negative. The optimal parameters suggest that RCD has better rate without these directions. See Thm 12 in suppl. material.

# 4 Experiments

## 4.1 Stochastic spectral coordinate descent (SSCD)

In our first experiment we study how the practical behavior of SSCD (Algorithm 4) depends on the choice of $k$. What we study here does not depend on the dimensionality of the problem ($n$), and hence it suffices to perform the experiments on small dimensional problems ($n = 30$). In this experiment we consider the regime of *clustered eigenvalues* described in the introduction and summarized in Table 3. In particular, we construct a synthetic matrix $\mathbf{A} \in \mathbb{R}^{30 \times 30}$ with the smallest 15 eigenvalues clustered in the interval $(5, 5 + \Delta)$ and the largest 15 eigenvalues clustered in the interval $(\theta, \theta + \Delta)$.

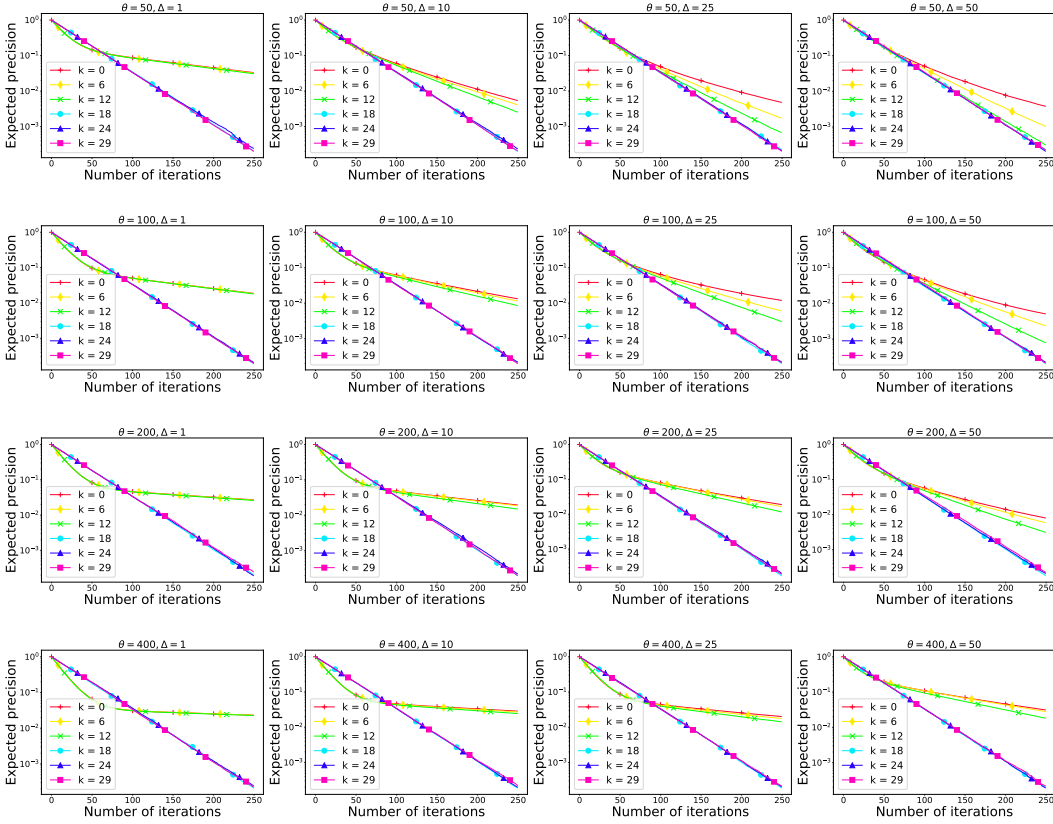

Figure 1: Expected precision $\mathbb{E}\left[\|x_t - x_*\|_{\mathbf{A}}^2 / \|x_0 - x_*\|_{\mathbf{A}}^2\right]$ versus # iterations of SSCD for symmetric positive definite matrices $\mathbf{A}$ of size $30 \times 30$ with different structures of spectra. The spectrum of $\mathbf{A}$ consists of 2 equally sized clusters of eigenvalues; one in the interval $(5, 5 + \Delta)$, and the other in the interval $(\theta, \theta + \Delta)$.

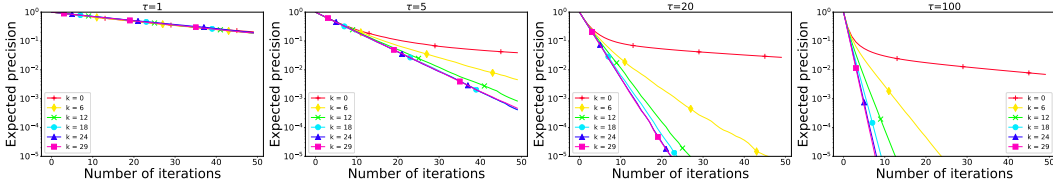

Figure 2: Expected precision versus # iterations of mini-batch SSCD for $\mathbf{A} \in \mathbb{R}^{30 \times 30}$ and several choices of mini-batch size $\tau$. The spectrum of $\mathbf{A}$ was chosen as a uniform discretization of the interval $[1, 60]$.

We vary the *tightness* parameter $\Delta$ and the *separation* parameter $\theta$, and study the performance of SSCD for various choices of $k$. See Figure 3.

Our first finding is a confirmation of the *phase transition* phenomenon predicted by our theory. Recall that the rate of SSCD (see Theorem 8) is $\tilde{\mathcal{O}}\left(\frac{(k+1)\lambda_{k+1} + \sum_{i=k+2}^{n} \lambda_i}{\lambda_{k+1}}\right)$. If $k < 15$, we know $\lambda_i \in (5, 5 + \Delta)$ for $i = 1, 2, \ldots, k + 1$, and $\lambda_i \in (\theta, \theta + \Delta)$ for $i = k + 2, \ldots, n$. Therefore, the rate can be estimated as $r_{small} := \tilde{\mathcal{O}}\left(k + 1 + (n - k - 1)(\theta + \Delta)/5\right)$. On the other hand, if $k \geq 15$, we know that $\lambda_i \in (\theta, \theta + \Delta)$ for $i = k + 1, \ldots, n$, and hence the rate can be estimated as $r_{large} := \tilde{\mathcal{O}}\left(k + 1 + (n - k - 1)(\theta + \Delta)/\theta\right)$. Note that if the separation $\theta$ between the two clusters is large, the rate $r_{large}$ is much better than the rate $r_{small}$. Indeed, in this regime, the rate $r_{large}$ becomes $\tilde{\mathcal{O}}(n)$, while $r_{small}$ can be arbitrarily large.

Going back to Figure 3, notice that this can be observed in the experiments. There is a clear *phase transition* at $k = 15$, as predicted be the above analysis. Methods using $k \in \{0, 6, 12\}$ are relatively slow (although still enjoying a linear rate), and tend to have similar behaviour, especially when $\Delta$

is small. On the other hand, methods using $k \in \{18, 24, 29\}$ are much faster, with a behaviour nearly independent of $\theta$ and $\Delta$. Moreover, as $\theta$ increases, the difference in the rates between the *slow* methods using $k \in \{0, 6, 12\}$ and the *fast* methods using $k \in \{18, 24, 29\}$ grows. We have performed more experiments with three clusters; see Fig 4 in the supplementary material.

## 4.2 Mini-batch SSCD

In Figure 2 we report on the behavior of mSSCD, the mini-batch version of SSCD, for four choices of the mini-batch parameter $\tau$, and several choices of $k$. Mini-batch of size $\tau$ is processed in parallel on $\tau$ processors, and the cost of a single iteration of mSSCD is (roughly) the same for all $\tau$. For $\tau = 1$, the method reduces to SSCD, considered in previous experiment (but on a different dataset). Since the number of iterations is small, there are no noticeable differences across using different values of $k$. As $\tau$ grows, however, all methods become faster. Mini-batching seems to be more useful as $k$ is larger. Moreover, we can observe that acceleration through mini-batching starts more aggressively for small values op $k$, and its added benefit for increasing values of $k$ is getting smaller and smaller. This means that even for relatively small values of $k$, mini-batching can be expected to lead to substantial speed-ups.

## 4.3 Matrix with 10 billion entries

In Figure 3 we report on an experiment using a synthetic problem with data matrix $\mathbf{A}$ of dimension $n = 10^5$ (i.e., potentially with $10^{10}$ entries). As all experiments were done on a laptop, we worked with sparse matrices with $10^6$ nonzeros only. In the first row of Figure 3 we consider matrix $\mathbf{A}$ with all eigenvalues distributed uniformly on the interval $[1, 100]$. We observe that SSCD with $k = 10^4$ (just 10% of $n$) requires about an *order of magnitude* less iterations than SSCD with $k = 0$ (=RCD). In the second row we consider a scenario where $l$ eigenvalues are small, contained in $[1, 2]$, with the rest of the eigenvalues contained in $[100, 200]$. We consider $l = 10$ and $l = 1000$ and study the behaviour of SSCD with $k = l$. We see that for $l = 10$, SSCD performs dramatically better than RCD: it is able to achieve machine precision while RCD struggles to reduce the initial error by a factor larger than $10^6$. For $l = 1000$, SSCD achieves error $10^{-9}$ while RCD struggles to push the error below $10^{-4}$. These tests show that in terms of # iterations, *SSCD has the capacity to accelerate on RCD by many orders of magnitude.*

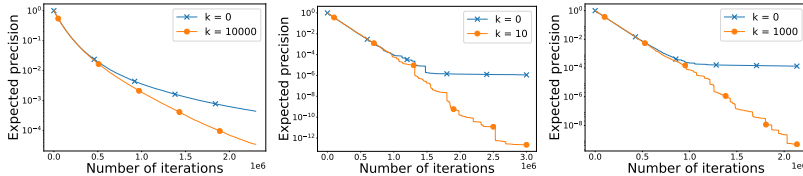

Figure 3: Expected precision $\mathbb{E}\left[\|x_t - x_*\|_{\mathbf{A}}^2 / \|x_0 - x_*\|_{\mathbf{A}}^2\right]$ versus # iterations of SSCD for a matrix $\mathbf{A} \in \mathbb{R}^{10^5 \times 10^5}$. Top row: spectrum of $\mathbf{A}$ is uniformly distributed on $[1, 100]$. Bottom row: spectrum contained in two clusters: $[1, 2]$ and $[100, 200]$.

## 5 Extensions

Our algorithms and convergence results can be extended to eigenvectors and conjugate directions which are only computed *approximately*. Some of this development can be found in the suppl. material (see Section E). Finally, as mentioned in the introduction, our results can be extended to the more general problem of minimizing $f(x) = \phi(\mathbf{A}x)$, where $\phi$ is smooth and strongly convex.

## Footnotes

[1]Many of our results can be extended to convex functions of the form $f(x) = \phi(\mathbf{A}x) - b^\top x$, where $\phi$ is a smooth and strongly convex function. However, due to space limitations, and the fact that we already have a lot to say in the special case $\phi(y) = \frac{1}{2}\|y\|^2$, we leave these more general developments to a follow-up paper.

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
