[Supplementary Material]

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

# Stochastic Spectral and Conjugate Descent Methods: Supplementary Material

## A  Mini-batch SD and SSCD

### A.1  Mini-batch SD

A mini-batch version of SD was developed by Richtárik and Takáč [18]. Here we restate the method as Algorithm 5.

---

**Algorithm 5** Mini-batch Stochastic Descent (mSD)

---

**Parameters:** Distribution $\mathcal{D}$; stepsize parameter $\omega > 0$; mini-batch size $\tau \geq 1$
**Initialize:** $x_0 \in \mathbb{R}^n$
**for** $t = 0, 1, 2, \ldots$ **do**
   **for** $i = 1, 2, \ldots, \tau$ **do**
      Sample $s_{ti} \sim \mathcal{D}$ and set $x_{t+1,i} = x_t - \omega \frac{s_{ti}^\top (\mathbf{A} x_t - b)}{s_{ti}^\top \mathbf{A} s_{ti}} s_{ti}$
   **end for**
   Set $x_{t+1} = \frac{1}{\tau} \sum_{i=1}^{\tau} x_{t+1,i}$
**end for**

---

**Lemma 9** (Convergence of mSD [18]). *Let $\mathcal{D}$ be proper with respect to $\mathbf{A}$, and let $0 < \omega < \frac{2}{\xi(\tau)}$, where $\xi(\tau) := \frac{1}{\tau} + \left(1 - \frac{1}{\tau}\right) \lambda_{\max}(\mathbf{W})$. Then*

$$\mathbb{E}[\|x_t - x_*\|_{\mathbf{A}}^2] \leq (\rho(\omega, \tau))^t \|x_0 - x_*\|_{\mathbf{A}}^2,$$

*where $\rho(\omega, \tau) = 1 - \omega[2 - \omega \xi(\tau)]\lambda_{\min}(\mathbf{W})$. For any fixed $\tau \geq 1$, the optimal stepsize choice is $\omega(\tau) = \frac{1}{\xi(\tau)}$ and the associated optimal rate is*

$$\rho(\omega(\tau), \tau) = 1 - \frac{\lambda_{\min}(\mathbf{W})}{\frac{1}{\tau} + \left(1 - \frac{1}{\tau}\right) \lambda_{\max}(\mathbf{W})}.$$

### A.2  Mini-batch SSCD

Specializing mSD to the distribution $\mathcal{D} = \mathcal{D}(\alpha, \beta_1, \ldots, \beta_k)$ gives rise to a new specific method which we call *mini-batch stochastic spectral coordinate descent (mSSCD)*, and formalize as Algorithm 6.

---

**Algorithm 6** Mini-batch Stochastic Spectral Coordinate Descent (mSSCD)

---

**Parameters:** Distribution $\mathcal{D}(\alpha, \beta_1, \ldots, \beta_k)$; relaxation parameter $\omega \in \mathbb{R}$; mini-batch size $\tau \geq 1$
**Initialize:** $x_0 \in \mathbb{R}^n$
**for** $t = 0, 1, 2, \ldots$ **do**
   **for** $i = 1, 2, \ldots, \tau$ **do**
      Sample $s_{ti} \sim \mathcal{D}(\alpha, \beta_1, \ldots, \beta_k)$ and set $x_{t+1,i} = x_t - \omega \frac{s_{ti}^\top (\mathbf{A} x_t - b)}{s_{ti}^\top \mathbf{A} s_{ti}} s_{ti}$
   **end for**
   Set $x_{t+1} = \frac{1}{\tau} \sum_{i=1}^{\tau} x_{t+1,i}$
**end for**

---

The rate of mSSCD is governed by the following result.

**Theorem 10.** *Consider mSSCD (Algorithm 6) for fixed $k \in \{0, 1, \ldots, n-1\}$ and optimal stepsize parameter $\omega(\tau) = \frac{1}{\xi(\tau)}$. The method converges linearly for all positive $\alpha > 0$ and nonnegative $\beta_i$. The best rate is obtained for parameters $\alpha = 1$ and $\beta_i = \lambda_{k+1} - \lambda_i$; and this is the unique choice of parameters leading to the best rate. In this case,*

$$\mathbb{E}[\|x_t - x_*\|_{\mathbf{A}}^2] \leq \left(1 - \frac{\lambda_{k+1}}{F_k}\right)^t \|x_0 - x_*\|_{\mathbf{A}}^2,$$

*where*

$$F_k := \frac{1}{\tau}\left((k+1)\lambda_{k+1} + \sum_{i=k+2}^{n}\lambda_i\right) + \left(1 - \frac{1}{\tau}\right)\lambda_n$$

. *Moreover, the rate improves as k grows, and we have*

$$\frac{\lambda_1}{\frac{1}{\tau}\mathrm{Tr}(\mathbf{A}) + \left(1 - \frac{1}{\tau}\right)\lambda_n} = \frac{\lambda_1}{F_0} \leq \cdots \leq \frac{\lambda_{k+1}}{F_k} \leq \cdots \leq \frac{\lambda_n}{F_{n-1}} = \frac{1}{\frac{n-1}{\tau}+1}.$$

If $k = 0$, mSSCD reduces to mini-batch RCD (with diagonal probabilities). Since $\frac{\lambda_1}{F_0} = \frac{\lambda_1}{\frac{1}{\tau}\mathrm{Tr}(\mathbf{A})+\left(1-\frac{1}{\tau}\right)\lambda_n}$, we recover the rate of mini-batch RCD [18]. With the choice $k = n - 1$ our method does *not* reduce to mSSD. However, the rates match.

# B  Extra Experiments

In this section we report on some additional experiments which shed more light on the behaviour of our methods.

## B.1  Performance on SSCD on A with three clusters eigenvalues

Figure 4: Expected precision $\mathbb{E}\left[\frac{||x_t-x_*||_{\mathbf{A}}^2}{||x_0-x_*||_{\mathbf{A}}^2}\right]$ versus the number of iterations of SSCD for symmetric positive definite matrices $\mathbf{A}$ of size $30 \times 30$ with different structures of spectrum. The spectrum of $\mathbf{A}$ consists of 3 equally sized clusters of eigenvalues; one in the interval $(10, 10 + \Delta)$, the second in the interval $(\theta, \theta + \Delta)$ and the third in the interval $(2\theta, 2\theta + \Delta)$. We show results for 16 combinations of $\theta$ and $\Delta$: $\Delta \in \{1, 10, 25, 100\}$ and $\theta \in \{100, 250, 500, 1000\}$.

In Figure 4 we report on experiments similar to those performed in Section 4, but on data matrix $\mathbf{A} \in \mathbb{R}^{30\times30}$ whose eigenvalues belong to three clusters, with 10 eigenvalues in each. We can observe

that the SSCD methods can be grouped into three categories: slow, fast, and very fast, depending on whether $k$ corresponds to the smallest 10 eigenvalues, the next cluster of 10 eigenvalues, or the 10 largest eigenvalues. That is, there are *two phase transitions*.

## B.2 Exponentially decaying eigenvalues

We now consider matrix $\mathbf{A} \in \mathbb{R}^{10 \times 10}$ with eigenvalues $2^0, 2^1, \ldots, 2^9$. We apply SSCD with increasing values of $k$ (see Figure 5).

Figure 5: Expected precision $\mathbb{E}\left[\frac{||x_t - x_*||_{\mathbf{A}}^2}{||x_0 - x_*||_{\mathbf{A}}^2}\right]$ versus the number of iterations of SSCD for symmetric positive definite matrix $\mathbf{A}$ of size $10 \times 10$.

We can see that the *performance boost accelerates as $k$ increases*. So, while one may not expect much speed-up for very small $k$, there will be substantial speed-up for moderate values of $k$. This is *predicted* by our theory. Indeed, consulting Table 3 (last column), we have $\alpha = 1/2$, and hence for $k = 0$ the theoretical rate is $\tilde{\mathcal{O}}(\frac{1}{\alpha^9})$. For general $k$ we have $\tilde{\mathcal{O}}(\frac{1}{\alpha^{9-k}})$. So, the speedup for value $k > 0$ compared to the baseline case of $k = 0$ (=RCD) is $2^k$, i.e., *exponential*.

# C Proofs

In this section we provide proofs of the statements from the main body of the paper. Table 4 provides a guide on where the proof of the various results can be found.

| Result | Section |
|---|---|
| Lemma 1 | C.1 |
| Theorem 2 | C.2 |
| Theorem 3 | C.3 |
| Theorem 4 | C.4 |
| Theorem 5 | C.5 |
| Theorem 6 | C.6 |
| Theorem 7 | C.7 |
| Theorem 8 | C.8 |
| Lemma 9 | C.9 |
| Theorem 10 | C.10 |

Table 4: Proof of lemmas and theorems stated in the main paper.

## C.1 Proof of Lemma 1

The result follows from Theorem 4.8(i) in [18] with the choice $\mathbf{B} = \mathbf{A}$. Note that since $x_* = \mathbf{A}^{-1}b$ is the unique solution of $\mathbf{A}x = b$, it is equal to the projection of $x_0$ onto the solution space of $\mathbf{A}x = b$, as required by the assumption in Theorem 4.8(i). It only remains to check that Assumption 3.5 (exactness) in [18] holds. In view of Theorem 3.6(iv) in [18], it suffices to check that the nullspace of $\mathbb{E}[\mathbf{H}]$ is trivial. However, this is equivalent to the assumption in Lemma 1 that $\mathbb{E}[\mathbf{H}]$ be invertible.

Finally, observe that

$$
\begin{aligned}
\tfrac{1}{2}\|x - x_*\|_{\mathbf{A}}^2 &= \tfrac{1}{2}(x - x_*)^\top \mathbf{A}(x - x_*) &&= \tfrac{1}{2}x^\top \mathbf{A}x + \tfrac{1}{2}x_*^\top \mathbf{A}x_* - x^\top \mathbf{A}x_* \\
&= \tfrac{1}{2}x^\top \mathbf{A}x + \tfrac{1}{2}x_*^\top \mathbf{A}x_* - x^\top \mathbf{A}\mathbf{A}^{-1}b &&\overset{(1)}{=} \quad f(x) + \tfrac{1}{2}x_*^\top \mathbf{A}x_* \\
&= f(x) - f(x_*).
\end{aligned}
$$

## C.2 Proof of Theorem 2

We will break down the proof into three steps.

1. First, let us show that Algorithm 2 is indeed SSD, as described in (4), i.e., $x_{t+1} = x_t - \frac{s_t^\top (\mathbf{A}x_t - b)}{s_t^\top \mathbf{A}s_t} s_t$. We known that $s_t = u_i$ with probability $1/n$. Since $\mathbf{A}u_i = \lambda_i u_i$, and assuming that at iteration $t$ we have $s_t = u_i$, we get

$$
\begin{aligned}
x_{t+1} &= x_t - \frac{u_i^\top (\mathbf{A}x_t - b)}{u_i^\top \mathbf{A}u_i} u_i &&= x_t - \frac{u_i^\top (\mathbf{A}x_t - b)}{\lambda_i} u_i \\
&= x_t - \frac{\lambda_i u_i^\top x_t - u_i^\top b}{\lambda_i} u_i &&= x_t - \left( u_i^\top x_t - \frac{u_i^\top b}{\lambda_i} \right) u_i.
\end{aligned}
$$

2. We now need to argue that the assumption that $\mathbb{E}[\mathbf{H}]$ is invertible is satisfied.

$$
\mathbb{E}[\mathbf{H}] \overset{(8)}{=} \sum_{i=1}^n \frac{1}{n} \frac{u_i u_i^\top}{u_i^\top \mathbf{A}u_i} = \sum_{i=1}^n \frac{1}{n} \frac{u_i u_i^\top}{\lambda_i}. \tag{12}
$$

Since $\mathbb{E}[\mathbf{H}]$ has positive eigenvalues $1/(n\lambda_i)$, it is invertible.

3. Applying Lemma 1, we get

$$
(1 - \lambda_{\max}(\mathbf{W}))^t \mathbb{E}[\|x_0 - x_*\|_{\mathbf{A}}^2] \le \mathbb{E}[\|x_t - x_*\|_{\mathbf{A}}^2] \le (1 - \lambda_{\min}(\mathbf{A}))^t \mathbb{E}[\|x_0 - x_*\|_{\mathbf{A}}^2].
$$

It remains to show that $\lambda_{\min}(\mathbf{W}) = \lambda_{\max}(\mathbf{W}) = \frac{1}{n}$. In view of (12), and since $\mathbf{A}^{1/2}u_i = \sqrt{\lambda_i}u_i$, we get

$$
\mathbf{W} \overset{(9)}{=} \mathbf{A}^{1/2} \mathbb{E}[\mathbf{H}] \mathbf{A}^{1/2} \overset{(12)}{=} \mathbf{A}^{1/2} \sum_{i=1}^n \frac{1}{n} \frac{u_i u_i^\top}{\lambda_i} \mathbf{A}^{1/2} = \sum_{i=1}^n \frac{1}{n} \frac{\mathbf{A}^{1/2} u_i u_i^\top \mathbf{A}^{1/2}}{\lambda_i} = \frac{1}{n} \mathbf{I}.
$$

## C.3 Proof of Theorem 3

Let $\mathbf{A}$ be a $2 \times 2$ symmetric positive definite matrix:

$$
\mathbf{A} = \begin{pmatrix} a & c \\ c & b \end{pmatrix}.
$$

We know that $a, b > 0$, and $ab - c^2 > 0$. Assume that $s_t = e_1 = (1, 0)^\top$ with probability $p > 0$ and $s_t = e_2 = (0, 1)^\top$ with probability $q > 0$, where $p + q = 1$. Then

$$
\mathbb{E}[\mathbf{H}] \overset{(8)}{=} p \frac{e_1 e_1^\top}{e_1^\top \mathbf{A}e_1} + q \frac{e_2 e_2^\top}{e_2^\top \mathbf{A}e_2} = \begin{pmatrix} \frac{p}{a} & 0 \\ 0 & \frac{q}{b} \end{pmatrix},
$$

and therefore,

$$
\mathbb{E}[\mathbf{H}]\mathbf{A} = \begin{pmatrix} p & p\frac{c}{a} \\ q\frac{c}{b} & q \end{pmatrix}.
$$

Note that $\mathbb{E}[\mathbf{H}]\mathbf{A}$ has the same eigenvalues as $\mathbf{W} = \mathbf{A}^{1/2}\mathbb{E}[\mathbf{H}]\mathbf{A}^{1/2}$. We now find the eigenvalues of $\mathbb{E}[\mathbf{H}]\mathbf{A}$ by finding the zeros of the characteristic polynomial:

$$
\det(\mathbb{E}[\mathbf{H}]\mathbf{A} - \lambda \mathbf{I}) = \det \begin{pmatrix} p - \lambda & p\frac{c}{a} \\ q\frac{c}{b} & q - \lambda \end{pmatrix} = \lambda^2 - \lambda + pq\left(1 - \frac{c^2}{ab}\right) = 0
$$

It can be seen that

$$
\lambda_{\min}(\mathbb{E}[\mathbf{H}]\mathbf{A}) = \frac{1}{2} - \frac{1}{2}\sqrt{1 - 4pq\left(1 - \frac{c^2}{ab}\right)} = \frac{1}{2} - \frac{1}{2}\sqrt{1 - 4p(1-p)\left(1 - \frac{c^2}{ab}\right)}.
$$

The expression $\lambda_{\min}(\mathbb{E}[\mathbf{H}]\mathbf{A})$ is maximized for $p = \frac{1}{2}$, independently of the values of $a, b$ and $c$.

## C.4 Proof of Theorem 4

Fix $n \geq 2$, and let $\Delta_n^+ := \{p \in \mathbb{R}^n \; : \; p > 0, \; \sum_i p_i = 1\}$ be the (interior of the) probability simplex. Further, let $\mathbf{A} = \mathrm{Diag}(\mathbf{A}_{11}, \mathbf{A}_{22}, \ldots, \mathbf{A}_{nn})$ be a diagonal matrix with positive diagonal entries.

The rate of RCD with any probabilities arises as a special case of Lemma 1. We therefore need to study the smallest eigenvalue of $\mathbf{W}$ (defined in (9)) as a function of $p = (p_1, \ldots, p_n)$. We have

$$\mathbf{H}(p) := \mathbb{E}_{s \sim \mathcal{D}}[\mathbf{H}] \overset{(8)}{=} \sum_i \frac{p_i}{\mathbf{A}_{ii}} e_i e_i^\top = \mathrm{Diag}(p_1/\mathbf{A}_{11}, p_2/\mathbf{A}_{22}, \ldots, p_n/\mathbf{A}_{nn}),$$

and hence

$$\mathbf{W} \overset{(9)}{=} \mathbf{W}(p) := \mathbf{A}^{1/2} \mathbf{H}(p) \mathbf{A}^{1/2} = \sum_{i=1}^n p_i e_i e_i^\top = \begin{pmatrix} p_1 & 0 & \cdots & \\ 0 & p_2 & \cdots & \\ \cdots & \cdots & \ddots & \\ 0 & 0 & \cdots & p_n \end{pmatrix}. \tag{13}$$

Note that $\lambda_{\min}(\mathbf{W}(p)) \overset{(13)}{=} \lambda_{\min}(\mathrm{Diag}(p_1, p_2, \ldots, p_n)) = \min_i p_i$, and thus

$$\max_{p \in \Delta_n^+} \lambda_{\min}(\mathbf{W}(p)) = \frac{1}{n}.$$

Clearly, the optimal probabilities are uniform: $p_i^* = \frac{1}{n}$ for all $i$.

## C.5 Proof of Theorem 5

We continue from the proof of Theorem 4.

1. Consider probabilities proportional to the diagonal elements: $p_i = \mathbf{A}_{ii}/\mathrm{Tr}(\mathbf{A})$ for all $i$. Choose $\mathbf{A}_{11} := t$, and $\mathbf{A}_{22} = \cdots = \mathbf{A}_{nn} = 1$. Then

$$\lambda_{\min}(\mathbf{W}(p)) \leq p_2 = \frac{\mathbf{A}_{22}}{\mathrm{Tr}(\mathbf{A})} = \frac{1}{t + n - 1} \longrightarrow 0 \text{ as } t \longrightarrow \infty.$$

2. Consider probabilities proportional to the squared row norms: $p_i = \|\mathbf{A}_{i:}\|^2/\mathrm{Tr}(\mathbf{A}^\top \mathbf{A})$ for all $i$. Choose $\mathbf{A}_{11} := t$, and $\mathbf{A}_{22} = \cdots = \mathbf{A}_{nn} = 1$. Then

$$\lambda_{\min}(\mathbf{W}(p)) \leq p_2 = \frac{\mathbf{A}_{22}}{\mathrm{Tr}(\mathbf{A}^\top \mathbf{A})} = \frac{1}{t^2 + n - 1} \longrightarrow 0 \text{ as } t \longrightarrow \infty.$$

In both cases, $\frac{\lambda_{\min}(\mathbf{W}(p))}{\lambda_{\min}(\mathbf{W}(p^*))}$ can be made arbitrarily small by a suitable choice of $t$.

## C.6 Proof of Theorem 6

The rate of RCD with any probabilities arises as a special case of Lemma 1. We therefore need to study the smallest eigenvalue of $\mathbf{W}$ (defined in (9)). Since we wish to show that the rate can be bad, we will first prove a lemma bounding $\lambda_{\min}(\mathbf{W})$ from above.

**Lemma 11.** *Let $0 < \lambda_1 \leq \lambda_2 \leq \cdots \leq \lambda_n$ be the eigenvalues of $\mathbf{A}$. Then*

$$\lambda_{\min}(\mathbf{W}) \leq \frac{1}{n}\left(\prod_{k=1}^n \frac{\lambda_k}{\mathbf{A}_{kk}}\right)^{1/n}. \tag{14}$$

*Proof.* We have

$$\mathbf{W} \overset{(9)}{=} \mathbf{A}^{\frac{1}{2}} \mathbb{E}[\mathbf{H}] \mathbf{A}^{\frac{1}{2}} \overset{(8)}{=} \mathbf{A}^{\frac{1}{2}}\left(\sum_{k=1}^n \frac{p_k e_k e_k^\top}{\mathbf{A}_{kk}}\right) \mathbf{A}^{\frac{1}{2}} = \mathbf{A}^{\frac{1}{2}} \mathrm{Diag}\left(\frac{p_k}{\mathbf{A}_{kk}}\right) \mathbf{A}^{\frac{1}{2}}.$$

From the above we see that the determinant of $\mathbf{W}$ is given by

$$\det(\mathbf{W}) = \det(\mathbf{A}) \prod_{k=1}^n \frac{p_k}{\mathbf{A}_{kk}}. \tag{15}$$

On the other hand, we have the trivial bound

$$\det(\mathbf{W}) = \prod_{k=1}^{n} \lambda_k(\mathbf{W}) \geq (\lambda_{\min}(\mathbf{W}))^n. \tag{16}$$

Putting these together, we get an upper bound on $\lambda_{\min}(\mathbf{W})$ in terms of the eigenvalues and diagonal elements of $\mathbf{A}$:

$$
\begin{aligned}
\lambda_{\min}(\mathbf{W}) &\overset{(16)}{\leq} \sqrt[n]{\det(\mathbf{W})} \overset{(15)}{=} \sqrt[n]{\det(\mathbf{A})} \cdot \sqrt[n]{\prod_{k=1}^{n} \frac{p_k}{\mathbf{A}_{kk}}} \\
&= \sqrt[n]{\det(\mathbf{A})} \cdot \sqrt[n]{\prod_{k=1}^{n} \frac{1}{\mathbf{A}_{kk}}} \cdot \sqrt[n]{\prod_{k=1}^{n} p_k} \\
&\overset{(*)}{\leq} \sqrt[n]{\det(\mathbf{A})} \cdot \sqrt[n]{\prod_{k=1}^{n} \frac{1}{\mathbf{A}_{kk}}} \cdot \frac{\sum_{k=1}^{n} p_k}{n} \\
&= \frac{\sqrt[n]{\det(\mathbf{A})}}{n} \cdot \sqrt[n]{\prod_{k=1}^{n} \frac{1}{\mathbf{A}_{kk}}} \\
&\overset{(16)}{=} \frac{1}{n} \sqrt[n]{\prod_{k=1}^{n} \frac{\lambda_k}{\mathbf{A}_{kk}}},
\end{aligned}
$$

where (*) follows from the arithmetic-geometric mean inequality. $\qquad\square$

**The Proof:** Let $\lambda_1, \ldots, \lambda_n$ are any positive real numbers. We now construct matrix $\mathbf{A} = \mathbf{M}\Lambda\mathbf{M}^\top$, where $\Lambda := \mathrm{Diag}(\lambda_1, \ldots, \lambda_n)$ and

$$
\mathbf{M} := \begin{pmatrix}
1/\sqrt{2} & 1/\sqrt{2} & 0 & \cdots & 0 \\
-1/\sqrt{2} & 1/\sqrt{2} & 0 & \cdots & 0 \\
0 & 0 & 1 & \cdots & 0 \\
\vdots & \vdots & \vdots & \ddots & 0 \\
0 & 0 & 0 & \cdots & 1
\end{pmatrix} \in \mathbb{R}^{n \times n}.
$$

Clearly, $\mathbf{A}$ is symmetric. Since $\mathbf{M}$ is orthonormal, $\lambda_1, \ldots, \lambda_n$ are, by construction, the eigenvalues of $\mathbf{A}$. Hence, $\mathbf{A}$ is symmetric and positive definite. Further, note that the diagonal entries of $\mathbf{A}$ are related to its eigenvalues as follows:

$$\mathbf{A}_{kk} = \begin{cases} \frac{\lambda_1 + \lambda_2}{2}, & k = 1, 2; \\ \lambda_k, & \text{otherwise.} \end{cases} \tag{17}$$

Applying Lemma 11, we get the bound

$$
\begin{aligned}
\lambda_{\min}(\mathbf{W}) &\overset{(14)}{\leq} \frac{1}{n} \left( \prod_{k=1}^{n} \frac{\lambda_k}{\mathbf{A}_{kk}} \right)^{1/n} \\
&= \frac{1}{n} \left( \prod_{k=1}^{2} \frac{\lambda_k}{\mathbf{A}_{kk}} \cdot \prod_{k=3}^{n} \frac{\lambda_k}{\mathbf{A}_{kk}} \right)^{1/n} \\
&\overset{(17)}{=} \frac{1}{n} \left( \prod_{k=1}^{2} \frac{\lambda_k}{\mathbf{A}_{kk}} \right)^{1/n} \\
&\overset{(17)}{=} \frac{1}{n} \left( \frac{4\lambda_1 \lambda_2}{(\lambda_1 + \lambda_2)^2} \right)^{1/n}.
\end{aligned}
$$

Let $c > 0$ be such that $\lambda_1 = c\lambda_2$. Then $\frac{4\lambda_1\lambda_2}{(\lambda_1+\lambda_2)^2} = \frac{4c}{(1+c)^2}$. If choose $c$ small enough so that $\frac{4c}{(1+c)^2} \leq \left(\frac{n}{T}\right)^n$, then $\lambda_{\min}(\mathbf{W}) \leq \frac{1}{T}$. The statement of the theorem follows.

## C.7 Proof of Theorem 7

Let $\mathbf{W} = \mathbf{U}\mathbf{\Lambda}\mathbf{U}^\top$ be the eigenvalue decomposition of $\mathbf{W}$, where $\mathbf{U} = [u_1, \ldots, u_n]$ are the eigenvectors, $\lambda_1(\mathbf{W}) \leq \ldots \leq \lambda_n(\mathbf{W})$ are the eigenvalues and $\mathbf{\Lambda} = \mathrm{Diag}\,(\lambda_1(\mathbf{W}), \ldots, \lambda_n(\mathbf{W}))$. From Theorem 4.3 of [18] we get

$$\mathbb{E}\left[\mathbf{U}^\top \mathbf{A}^{1/2}(x_t - x_*)\right] = (\mathbf{I} - \mathbf{\Lambda})^t \mathbf{U}^\top \mathbf{A}^{1/2}(x_0 - x_*). \tag{18}$$

Now we use Jensen's inequality and get

$$\mathbb{E}\left[\|x_t - x_*\|_{\mathbf{A}}^2\right] = \mathbb{E}\left[\left\|\mathbf{U}^\top \mathbf{A}^{1/2}(x_t - x_*)\right\|_2^2\right] \geq$$

$$\geq \left\|\mathbb{E}\left[\mathbf{U}^\top \mathbf{A}^{1/2}(x_t - x_*)\right]\right\|_2^2 \overset{(18)}{=} \left\|(\mathbf{I} - \mathbf{\Lambda})^t \mathbf{U}^\top \mathbf{A}^{1/2}(x_0 - x_*)\right\|_2^2 =$$

$$= \sum_{i=1}^n (1 - \lambda_i(\mathbf{W}))^{2t}\left(u_i^\top \mathbf{A}^{1/2}(x_0 - x_*)\right)^2 \geq (1 - \lambda_1(\mathbf{W}))^{2t}\left(u_1^\top \mathbf{A}^{1/2}(x_0 - x_*)\right)^2.$$

Now we take an example of matrix $\mathbf{A}$, for which we set $\lambda_{\min}(\mathbf{W}) \leq \frac{1}{T}$ for arbitrary $T > 0$, like we did in Section C.6. We also choose $x_0 = x_* + \mathbf{A}^{-1/2}u_1$. For this choice of $\mathbf{A}$ and $x_0$ we get $\|x_0 - x_*\|_{\mathbf{A}}^2 = \|u_1\|_2^2$ and

$$\mathbb{E}\left[\|x_t - x_*\|_{\mathbf{A}}^2\right] \geq (1 - \lambda_1(\mathbf{W}))^{2t} \|u_1\|_2^2 \geq \left(1 - \frac{1}{T}\right)^{2t} \|u_1\|_2^2 = \left(1 - \frac{1}{T}\right)^{2t} \|x_0 - x_*\|_{\mathbf{A}}^2.$$

## C.8 Proof of Theorem 8

We divide the proof into several steps.

1. Let us first show that SSCD converges with a linear rate for any choice of $\alpha > 0$ and nonnegative $\{\beta_i\}$. Since SSCD arises as a special case of SD, it suffices to apply Lemma 1. In order to apply this lemma, we need to argue that $\mathcal{D} = \mathcal{D}(\alpha, \beta_1, \ldots, \beta_n)$ is a proper distribution. Indeed,

$$\mathbb{E}_{s \sim \mathcal{D}}[\mathbf{H}] \overset{(8)}{=} \sum_{i=1}^n p_i \frac{e_i e_i^\top}{e_i^\top \mathbf{A} e_i} + \sum_{i=1}^k p_{n+i} \frac{u_i u_i^\top}{u_i^\top \mathbf{A} u_i}$$

$$= \frac{1}{C_k}\left(\alpha \mathbf{I} + \sum_{i=1}^k u_i u_i^\top \frac{\beta_i}{\lambda_i}\right) \tag{22}$$

$$\succeq \frac{\alpha}{C_k}\mathbf{I} \quad \succ \quad 0.$$

2. For the specific choice of parameters $\alpha = 1$ and $\beta_i = \lambda_{k+1} - \lambda_i$ we have

$$\mathbb{E}_{s \sim \mathcal{D}}[\mathbf{H}] = \frac{1}{C_k}\left(\mathbf{I} + \sum_{i=1}^k u_i u_i^\top \frac{\lambda_{k+1} - \lambda_i}{\lambda_i}\right),$$

and $C_k = (k+1)\lambda_{k+1} + \sum_{i=k+2}^m \lambda_i$. Therefore,

$$\mathbb{E}_{s \sim \mathcal{D}}[\mathbf{A}\mathbf{H}] = \frac{1}{C_k}\left(\sum_{i=1}^k \lambda_{k+1} u_i u_i^\top + \sum_{i=k+1}^n \lambda_i u_i u_i^\top\right).$$

The minimal eigenvalue of this matrix, which has the same spectrum as $\mathbf{W}$, is

$$\lambda_{\min}(\mathbb{E}_{s \sim \mathcal{D}}[\mathbf{A}\mathbf{H}]) = \frac{\lambda_{k+1}}{C_k} = \frac{\lambda_{k+1}}{(k+1)\lambda_{k+1} + \sum_{i=k+2}^n \lambda_i}.$$

The main statement follows by applying Lemma 1.

3. We now show that the rate improves as $k$ increases. Indeed,

$$k + \frac{1}{\lambda_{k+1}} \sum_{i=k+1}^{m} \lambda_i = k + 1 + \frac{1}{\lambda_{k+1}} \sum_{i=k+2}^{m} \lambda_i \geq k + 1 + \frac{1}{\lambda_{k+2}} \sum_{i=k+2}^{m} \lambda_i.$$

By taking reciprocals, we get

$$\frac{\lambda_{k+2}}{(k+1)\lambda_{k+2} + \sum\limits_{i=k+2}^{m} \lambda_i} \geq \frac{\lambda_{k+1}}{k\lambda_{k+1} + \sum\limits_{i=k+1}^{m} \lambda_i}.$$

4. It remains to establish optimality of the specific parameter choice $\alpha = 1$ and $\beta_i = \lambda_{k+1} - \lambda_i$. Continuing from (22), we get

$$\mathbb{E}_{s \sim \mathcal{D}}[\mathbf{AH}] \overset{(22)}{=} \frac{1}{C_k} \left( \sum_{i=1}^{n} u_i u_i^\top \alpha \lambda_i + \sum_{i=1}^{k} u_i u_i^\top \beta_i \right) =$$

$$= \frac{1}{C_k} \left( \sum_{i=1}^{k} (\alpha \lambda_i + \beta_i) u_i u_i^\top + \sum_{i=k+1}^{n} \alpha \lambda_i u_i u_i^\top \right).$$

The eigenvalues of $\mathbb{E}_{s \sim \mathcal{D}}[\mathbf{AH}]$ are $\{\frac{\alpha \lambda_i + \beta_i}{C_k}\}_{i=1}^{k} \cup \{\frac{\alpha \lambda_i}{C_k}\}_{i=k+1}^{n}$. Let $\gamma$ be the smallest eigenvalue, i.e., $\gamma := \lambda_{\min}(\mathbb{E}_{s \sim \mathcal{D}}[\mathbf{AH}]) = \frac{\theta}{C_k}$, and $\Omega$ be the largest eigenvalue, i.e., $\Omega := \lambda_{\max}(\mathbb{E}_{s \sim \mathcal{D}}[\mathbf{AH}]) = \frac{\Delta}{C_k}$, where $\theta$ and $\Delta$ are appropriate constants. There are now two options.

(a) $\gamma = \frac{\alpha \lambda_{k+1}}{C_k}$. Then $\alpha \lambda_i + \beta_i \geq \alpha \lambda_{k+1}$ for $i \in \{1, \ldots, k\}$. In this case we obtain:

$$C_k = \alpha \mathrm{Tr}(\mathbf{A}) + \sum_{i=1}^{k} \beta_i = \sum_{i=1}^{k} (\alpha \lambda_i + \beta_i) + \alpha \sum_{i=k+1}^{n} \lambda_i \geq \alpha \left( k\lambda_{k+1} + \sum_{i=k+1}^{n} \lambda_i \right)$$

and therefore

$$\gamma \leq \frac{\lambda_{k+1}}{k\lambda_{k+1} + \sum\limits_{i=k+1}^{n} \lambda_i}.$$

(b) $\gamma = \frac{\alpha \lambda_j + \beta_j}{C_k} = \frac{\theta}{C_k}$ for some $j \in \{1, \ldots, k\}$. Then

$$C_k = \alpha \mathrm{Tr}(\mathbf{A}) + \sum_{i=1}^{k} \beta_i = \sum_{i=1}^{k} (\alpha \lambda_i + \beta_i) + \alpha \sum_{i=k+1}^{n} \lambda_i \geq k\theta + \alpha \sum_{i=k+1}^{n} \lambda_i$$

whence

$$\gamma \leq \frac{\theta}{k\theta + \alpha \sum\limits_{i=k+1}^{n} \lambda_i}.$$

Note that the function $f(\theta) = \frac{\theta}{k\theta + \alpha \sum\limits_{i=k+1}^{n} \lambda_i}$ increases monotonically:

$$f'(\theta) = \frac{1}{k\theta + \alpha \sum\limits_{i=k+1}^{n} \lambda_i} - \frac{k\theta}{(k\theta + \alpha \sum\limits_{i=k+1}^{n} \lambda_i)^2} = \frac{\alpha \sum\limits_{i=k+1}^{n} \lambda_i}{(k\theta + \alpha \sum\limits_{i=k+1}^{n} \lambda_i)^2} > 0.$$

From this and inequality $\alpha \lambda_{k+1} \geq \theta$ we get

$$\gamma \leq \frac{\alpha \lambda_{k+1}}{\alpha(k\lambda_{k+1} + \sum\limits_{i=k+1}^{n} \lambda_i)} = \frac{\lambda_{k+1}}{k\lambda_{k+1} + \sum\limits_{i=k+1}^{n} \lambda_i}.$$

In both possible cases we have shown that

$$\lambda_{\min}\left(\mathbb{E}_{s\sim\mathcal{D}}[\mathbf{AH}]\right) \leq \frac{\lambda_{k+1}}{k\lambda_{k+1} + \sum\limits_{i=k+1}^{n}\lambda_i}.$$

So, it is the optimal rate in this family of methods. Optimal distribution is unique and it is:

$$s\sim\mathcal{D} \quad\Leftrightarrow\quad s = \begin{cases} e_i & \text{with probability } p_i = \frac{\mathbf{A}_{ii}}{C_k} \quad i = 1,2,\ldots,n \\ u_i & \text{with probability } p_{n+i} = \frac{\lambda_{k+1}-\lambda_i}{C_k} \quad i = 1,2,\ldots,k, \end{cases}$$

where $C_k = k\lambda_{k+1} + \sum\limits_{i=k+1}^{n}\lambda_i$.

## C.9  Proof of Lemma 9

The steps are analogous to the proof of Lemma 1.

## C.10  Proof of Theorem 10

Let $C_k = (k+1)\lambda_{k+1} + \sum\limits_{i=k+2}^{n}\lambda_i$ $\gamma = \frac{\theta}{C_k}$ — the minimal eigenvalue of the matrix $\mathbf{W}$ and $\Omega = \frac{\Delta}{C_k}$ — the maximal eigenvalue of the matrix $\mathbf{W}$. The optimal rate of the method [18] is

$$r(\tau) = \frac{\gamma}{\frac{1}{\tau} + \left(1 - \frac{1}{\tau}\right)\Omega} = \frac{\theta}{\frac{1}{\tau}C_k + \left(1 - \frac{1}{\tau}\right)\Delta}.$$

From the Section C.8 we have

$$\mathbb{E}_{s\sim\mathcal{D}}[\mathbf{AH}] = \frac{1}{C_k}\left(\sum_{i=1}^{k}\lambda_{k+1}u_i u_i^\top + \sum_{i=k+1}^{n}\lambda_i u_i u_i^\top\right).$$

There are two options.

1. $\gamma = \frac{\alpha\lambda_{k+1}}{C_k}$. Then $\alpha\lambda_i + \beta_i \geq \alpha\lambda_{k+1}$ for $i \in \{1,\ldots,k\}$ and $\Delta \geqslant \alpha\lambda_n$. In this case we obtain:

$$C_k = \alpha\mathrm{Tr}\,(\mathbf{A}) + \sum_{i=1}^{k}\beta_i = \sum_{i=1}^{k}(\alpha\lambda_i + \beta_i) + \alpha\sum_{i=k+1}^{n}\lambda_i \geq \alpha\left(k\lambda_{k+1} + \sum_{i=k+1}^{n}\lambda_i\right)$$

and therefore

$$r(\tau) \leq \frac{\alpha\lambda_{k+1}}{\frac{\alpha}{\tau}\left(k\lambda_{k+1} + \sum\limits_{i=k+1}^{n}\lambda_i\right) + \left(1 - \frac{1}{\tau}\right)\alpha\lambda_n} = \frac{\lambda_{k+1}}{\frac{1}{\tau}\left(k\lambda_{k+1} + \sum\limits_{i=k+1}^{n}\lambda_i\right) + \left(1 - \frac{1}{\tau}\right)\lambda_n}.$$

2. $\gamma = \frac{\alpha\lambda_j + \beta_j}{C_k} = \frac{\theta}{C_k}$ for some $j \in \{1,\ldots,k\}$. Then

$$C_k = \alpha\mathrm{Tr}\,(\mathbf{A}) + \sum_{i=1}^{k}\beta_i = \sum_{i=1}^{k}(\alpha\lambda_i + \beta_i) + \alpha\sum_{i=k+1}^{n}\lambda_i \geq k\theta + \alpha\sum_{i=k+1}^{n}\lambda_i, \quad \Delta \geq \alpha\lambda_n$$

whence

$$r(\tau) \leqslant \frac{\theta}{\frac{1}{\tau}\left(k\theta + \alpha\sum\limits_{i=k+1}^{n}\lambda_i\right) + \left(1 - \frac{1}{\tau}\right)\alpha\lambda_n}.$$

Note that the function $f(\theta) = \dfrac{\theta}{\frac{1}{\tau}\left(k\theta + \alpha \sum\limits_{i=k+1}^{n} \lambda_i\right) + \left(1 - \frac{1}{\tau}\right)\alpha\lambda_n}$ increases monotonically:

$$f'(\theta) = \frac{1}{\frac{1}{\tau}\left(k\theta + \alpha \sum\limits_{i=k+1}^{n} \lambda_i\right) + \left(1 - \frac{1}{\tau}\right)\alpha\lambda_n} - \frac{\frac{k}{\tau}\theta}{\left(\frac{1}{\tau}\left(k\theta + \alpha \sum\limits_{i=k+1}^{n} \lambda_i\right) + \left(1 - \frac{1}{\tau}\right)\alpha\lambda_n\right)^2}$$

$$= \frac{\frac{\alpha}{\tau}\sum\limits_{i=k+1}^{n}\lambda_i + \left(1 - \frac{1}{\tau}\right)\alpha\lambda_n}{\left(\frac{1}{\tau}\left(k\theta + \alpha \sum\limits_{i=k+1}^{n} \lambda_i\right) + \left(1 - \frac{1}{\tau}\right)\alpha\lambda_n\right)^2} > 0.$$

From this and inequality $\alpha\lambda_{k+1} \geq \theta$ we get

$$r(\tau) \leq \frac{\alpha\lambda_{k+1}}{\frac{1}{\tau}\left(\alpha k\lambda_{k+1} + \alpha \sum\limits_{i=k+1}^{n} \lambda_i\right) + \left(1 - \frac{1}{\tau}\right)\alpha\lambda_n} = \frac{\lambda_{k+1}}{\frac{1}{\tau}\left(k\lambda_{k+1} + \sum\limits_{i=k+1}^{n} \lambda_i\right) + \left(1 - \frac{1}{\tau}\right)\lambda_n}.$$

For both possible cases we shown that $r(\tau) \leq \dfrac{\lambda_{k+1}}{\frac{1}{\tau}\left(k\lambda_{k+1} + \sum\limits_{i=k+1}^{n} \lambda_i\right) + \left(1 - \frac{1}{\tau}\right)\lambda_n}$. So, it is the optimal rate in this family of methods. Note that $\alpha$ could be any positive number. Optimal distribution is unique and it is:

$$s \sim \mathcal{D} \quad \Leftrightarrow \quad s = \begin{cases} e_i & \text{with probability } p_i = \frac{\mathbf{A}_{ii}}{C_k} \quad i = 1, 2, \ldots, n \\ u_i & \text{with probability } p_{n+i} = \frac{\lambda_{k+1} - \lambda_i}{C_k} \quad i = 1, 2, \ldots, k, \end{cases}$$

where $C_k = k\lambda_{k+1} + \sum\limits_{i=k+1}^{n} \lambda_i$. For $k = 0$ we obtain mRCD, for $k = n - 1$ we get the optimal rate $\dfrac{\frac{1}{n}}{\frac{1}{\tau} + \left(1 - \frac{1}{\tau}\right)\frac{1}{n}}$ and rate increases when $k$ increases.

# D   Results mentioned informally in the paper

## D.1   Adding "largest" eigenvectors does not help

In Section 3 describing the SSCD method we have argued, without supplying any detail, that it does not make sense to consider replacing the $k$ "smallest" eigenvectors with a few "largest" eigenvectors. Here we make this statement precise, and prove it.

Fix $k \in \{0, 1, \ldots, n-1\}$ and consider running stochastic descent with the distribution $\mathcal{D}$ defined via

$$s \sim \mathcal{D} \quad \Leftrightarrow \quad s = \begin{cases} e_i & \text{with probability } p_i = \frac{\alpha \mathbf{A}_{ii}}{C_k} \quad i = 1, 2, \ldots, n \\ u_i & \text{with probability } p_{n-k+i} = \frac{\beta_i}{C_k} \quad i = k+1, k+2, \ldots, n, \end{cases}$$

where $C_k = \alpha \mathrm{Tr}\left(\mathbf{A}\right) + \sum\limits_{i=k+1}^{n} \beta_i$ and for $\beta_i \geq 0$ for $i \in \{1, 2, \ldots, k\}$.

That is, we consider "enriching" RCD with a collection of a $n - k$ eigenvectors corresponding to the $n - k$ largest eigenvectors of $\mathbf{A}$. We have the following negative result, which loosely speaking says that it is not worth enriching RCD with such vectors.

**Theorem 12.** *The optimal parameters of the above method are $k = n$ or $\beta_i = 0$ for all $i = k+1, \ldots, n$.*

*Proof.* We follow similar steps as in the proof of Theorem 8. In this setting we have

$$\mathbb{E}_{s\sim\mathcal{D}}[\mathbf{H}] = \frac{1}{C_k}\left(\alpha\mathbf{I} + \sum\limits_{i=k+1}^{n} \frac{\beta_i}{\lambda_i} u_i u_i^\top\right),$$

whence

$$\mathbf{A}\mathbb{E}_{s\sim\mathcal{D}}[\mathbf{H}] = \frac{1}{C_k}\left(\alpha\mathbf{A} + \sum\limits_{i=k+1}^{n} \beta_i u_i u_i^\top\right) = \frac{1}{C_k}\left(\sum\limits_{i=1}^{k} \alpha\lambda_i u_i u_i^\top + \sum\limits_{i=k+1}^{n} (\beta_i + \alpha\lambda_i) u_i u_i^\top\right)$$

and

$$\lambda_{\min}\left(\mathbf{A}\mathbb{E}_{s\sim\mathcal{D}}[\mathbf{H}]\right) = \frac{\alpha\lambda_1}{C_k} \leq \frac{\alpha\lambda_1}{\alpha\operatorname{Tr}(\mathbf{A})} = \frac{\lambda_1}{\operatorname{Tr}(\mathbf{A})}.$$

It means that the best rate in this family of methods is obtained when $k = n$ or $\beta_i = 0$ for all $i = k+1, \ldots, n$. $\qquad\square$

So, to use spectral information about $n - k$ last eigenvectors we should use more complicated distributions (for instance, one may need to replace $\alpha$ by $\alpha_i$).

### D.2 Stochastic Conjugate Descent

The lemma below was referred to in Section 2.2. As explained in that section, this lemma can be used to argue that stochastic conjugate descent achieves the same rate as SSD: $\mathcal{O}(n\log\frac{1}{\epsilon})$.

**Lemma 13.** *Let $\{v_1 \ldots v_n\}$ be an $\mathbf{A}$-orthonormal system:*

$$v_i^\top \mathbf{A} v_j = \begin{cases} 1 & i = j \\ 0 & i \neq j \end{cases}.$$

*If distribution $\mathcal{D}$ consists of vectors $v_i$ chosen with uniform probabilities, then $\lambda_{\min}(\mathbf{W}) = \frac{1}{n}$*

*Proof.* That is,

$$\mathbf{W} = \mathbf{A}^{1/2}\mathbb{E}[\mathbf{H}]\mathbf{A}^{1/2} = \frac{1}{n}\sum_{i=1}^{n}\frac{\mathbf{A}^{1/2}v_i v_i^\top \mathbf{A}^{1/2}}{v_i^\top \mathbf{A} v_i} = \frac{1}{n}\sum_{i=1}^{n}\mathbf{A}^{1/2}v_i v_i^\top \mathbf{A}^{1/2}.$$

Making a substitution $u_i = \mathbf{A}^{1/2}s_i$, we get

$$\mathbf{W} = \frac{1}{n}\sum_{i=1}^{n}u_i u_i^\top = \frac{1}{n}\mathbf{I},$$

because $\{u_1 \ldots u_n\}$ is orthonormal system. $\qquad\square$

## E   Inexact Stochastic Conjuagate Descent

In Section 2.2 we stated, that we can achieve an optimal rate of stochastic descent by using uniform distribution over a set of $n$ $\mathbf{A}$-conjugate directions. In this section we consider the case when $\mathbf{A}$-conjugate directions are computed approximately.

More formally, we consider a system of vectors $v_1, \ldots, v_n$, which satisfies $\left|v_i^\top \mathbf{A} v_j\right| \leq \varepsilon$ for $i \neq j$ and $v_i^\top \mathbf{A} v_i = 1$ for some parameter $\varepsilon > 0$. Further we'll call such vectors $\varepsilon$-approximate $\mathbf{A}$-conjugate vectors.

Now we formalize the idea of using approximate $\mathbf{A}$-conjugate directions in Stochastic Conjugate Descent, which leads to Algorithm 7.

---
**Algorithm 7** Inexact Stochastic Conjugate Descent (iSconD)

---
**Initialize:** $x_0 \in \mathbb{R}^n$; $v_1, \ldots, v_n$: $\varepsilon$-approximate $\mathbf{A}$-conjugate directions
**for** $t = 0, 1, 2, \ldots$ **do**
    Choose $i \in [n]$ uniformly at random
    Set $x_{t+1} = x_t - v_i^\top \left(\mathbf{A}x_t - b\right)v_i$
**end for**

---

For this algorithm we are going to obtain rate $\mathcal{O}(n\log\frac{1}{\epsilon})$, the optimal rate for stochastic descent.

### E.1 Lemma

**Lemma 14.** *Let $\mathbf{S} = [v_1, \ldots, v_n]$, where $v_1, \ldots, v_n$ are $\varepsilon$-approximate $\mathbf{A}$-conjugate vectors. If $\varepsilon$ satisfies*

$$\varepsilon < \frac{1}{n-1} \tag{25}$$

*then $\tilde{\mathbf{I}} := \mathbf{S}^\top \mathbf{A} \mathbf{S}$ is positive definite matrix and*

$$\lambda_{\min}(\mathbf{A}^{1/2}\mathbf{S}\mathbf{S}^\top\mathbf{A}^{1/2}) \geq 1 - \varepsilon(n-1)\frac{1 + \varepsilon(n-1)}{1 - \varepsilon(n-1)} \tag{26}$$

$$\lambda_{\max}(\mathbf{A}^{1/2}\mathbf{S}\mathbf{S}^\top\mathbf{A}^{1/2}) \leq 1 + \varepsilon(n-1)\frac{1 + \varepsilon(n-1)}{1 - \varepsilon(n-1)}$$

*Proof.* For unit vector $x$ we can write

$$x^\top \tilde{\mathbf{I}} x = \sum_{i,l} x_i x_l \tilde{\mathbf{I}}_{il} = 1 + \sum_{i,l:i\neq l} x_i x_l \tilde{\mathbf{I}}_{il} \geq 1 - \varepsilon \sum_{i,l:i\neq l} \frac{1}{2}(x_i^2 + x_l^2) = 1 - \varepsilon(n-1).$$

Under condition (25) we get $x^\top \tilde{\mathbf{I}} x > 0$ for any $x$, which proves the first part of lemma.

Since $\mathbf{S}^\top \mathbf{A} \mathbf{S}$ is positive definite, vectors $\mathbf{A}^{1/2}v_1, \ldots, \mathbf{A}^{1/2}v_n$ are linearly independent. Any unit vector $x$ may be represented as $x = \mathbf{A}^{1/2}\mathbf{S}\alpha$ with normalization condition:

$$1 = x^\top x = \alpha^\top \tilde{\mathbf{I}}\alpha = \alpha^\top\alpha + \sum_{i,l:i\neq l} \tilde{\mathbf{I}}_{il}\alpha_i\alpha_l,$$

or

$$\alpha^\top\alpha = 1 - \sum_{i,l:i\neq l} \tilde{\mathbf{I}}_{il}\alpha_i\alpha_l. \tag{28}$$

Now we can analyse spectrum of matrix $\mathbf{A}^{1/2}\mathbf{S}\mathbf{S}^\top\mathbf{A}^{1/2}$.

$$x^\top \mathbf{A}^{1/2}\mathbf{S}\mathbf{S}^\top\mathbf{A}^{1/2}x = \alpha^\top \mathbf{S}^\top \mathbf{A}\mathbf{S}\mathbf{S}^\top\mathbf{A}\mathbf{S}\alpha = \alpha^\top \tilde{\mathbf{I}}^2\alpha = \left\|\tilde{\mathbf{I}}\alpha\right\|_2^2 = \sum_{i=1}^n \left(\sum_{l=1}^n \tilde{\mathbf{I}}_{il}\alpha_l\right)^2 =$$

$$= \sum_{i=1}^n \left(\alpha_i + \sum_{l:l\neq i} \tilde{\mathbf{I}}_{il}\alpha_l\right)^2 = \sum_{i=1}^n \left(\alpha_i^2 + 2\alpha_i \sum_{l:l\neq i} \tilde{\mathbf{I}}_{il}\alpha_l + \left(\sum_{l:l\neq i} \tilde{\mathbf{I}}_{il}\alpha_l\right)^2\right).$$

Using (28) we get

$$x^\top \mathbf{A}^{1/2}\mathbf{S}\mathbf{S}^\top\mathbf{A}^{1/2}x = 1 + \underbrace{\sum_{i,l:l\neq i} \tilde{\mathbf{I}}_{il}\alpha_i\alpha_l}_{R_1} + \underbrace{\sum_{i=1}^n \left(\sum_{l:l\neq i} \tilde{\mathbf{I}}_{il}\alpha_l\right)^2}_{R_2} = 1 + R_1 + R_2 \tag{29}$$

To estimate $|R_1|$ and $|R_2|$ we need to estimate $\alpha^\top\alpha$ using (28):

$$\alpha^\top\alpha \leq 1 + \varepsilon \sum_{i,l:i\neq l} \frac{\alpha_i^2 + \alpha_l^2}{2} = 1 + \varepsilon(n-1)\alpha^\top\alpha,$$

which under condition (25) implies that $\alpha^\top\alpha \leq \frac{1}{1-\varepsilon(n-1)}$. Now we can estimate $|R_1|$ and $|R_2|$.

$$R_1 \leq \varepsilon \sum_{i,l:i\neq l} \frac{\alpha_i^2 + \alpha_l^2}{2} = \varepsilon(n-1)\alpha^\top\alpha \leq \frac{\varepsilon(n-1)}{1 - \varepsilon(n-1)} \tag{30}$$

$$R_2 \leq \sum_{i=1}^n (n-1)\sum_{l:l\neq i} \alpha_l^2\varepsilon^2 = \varepsilon^2(n-1)^2\alpha^\top\alpha \leq \frac{\varepsilon^2(n-1)^2}{1 - \varepsilon(n-1)} \tag{31}$$

Finally from (29), (30) and (31) we get

$$\lambda_{\min}(\mathbf{A}^{1/2}\mathbf{S}\mathbf{S}^\top\mathbf{A}^{1/2}) \geq 1 - \frac{\varepsilon(n-1) + \varepsilon^2(n-1)^2}{1 - \varepsilon(n-1)} = 1 - \varepsilon(n-1)\frac{1 + \varepsilon(n-1)}{1 - \varepsilon(n-1)}$$

$$\lambda_{\max}(\mathbf{A}^{1/2}\mathbf{S}\mathbf{S}^\top\mathbf{A}^{1/2}) \leq 1 + \frac{\varepsilon(n-1) + \varepsilon^2(n-1)^2}{1 - \varepsilon(n-1)} = 1 + \varepsilon(n-1)\frac{1 + \varepsilon(n-1)}{1 - \varepsilon(n-1)}$$

$\square$

**Corollary 14.1.** *If* $\varepsilon < \frac{\sqrt{2}-1}{(n-1)}$ *then* $\lambda_{\min}(\mathbf{A}^{1/2}\mathbf{S}\mathbf{S}^\top\mathbf{A}^{1/2}) > 0$ *and condition number of* $\mathbf{A}^{1/2}\mathbf{S}\mathbf{S}^\top\mathbf{A}^{1/2}$ *has the following bound:*

$$\frac{\lambda_{\max}(\mathbf{A}^{1/2}\mathbf{S}\mathbf{S}^\top\mathbf{A}^{1/2})}{\lambda_{\min}(\mathbf{A}^{1/2}\mathbf{S}\mathbf{S}^\top\mathbf{A}^{1/2})} < \frac{1 + \varepsilon^2(n-1)^2}{1 - 2\varepsilon(n-1) - \varepsilon^2(n-1)^2}$$

## E.2 Rate of convergence

The following theorem gives the rate of convergence of iSconD.

**Theorem 15.** *Let* $\mathbf{S} = [v_1 \ldots v_n]$, *where* $\{v_1 \ldots v_n\}$ *is* $\varepsilon$-*approximate* $\mathbf{A}$-*conjugate system. If* $\varepsilon \leq \frac{1}{3(n-1)}$ *then* $\lambda_{\min}(\mathbf{W}) > \frac{1}{3n}$, *which means that the rate of iSconD is* $\mathcal{O}(n \log \frac{1}{\epsilon})$.

*Proof.* As in Lemma 13, we can show that $\mathbf{W} = \frac{1}{n}\mathbf{A}^{1/2}\mathbf{S}\mathbf{S}^\top\mathbf{A}$, where $\mathbf{S} = [v_1 \ldots v_n]$. Using bound (26) and Corollary 14.1, we get

$$\lambda_{\min}(\mathbf{W}) > \frac{1}{n}\left(1 - \varepsilon(n-1)\frac{1 + \varepsilon(n-1)}{1 - \varepsilon(n-1)}\right)$$

for small enough $\varepsilon$ (see Corollary 14.1). For $\varepsilon = \frac{1}{3(n-1)}$ we get $\lambda_{\min}(\mathbf{W}) > \frac{1}{3n}$. $\square$

## E.3 Experiment

Figure 6 illustrates the theoretical results about iSonD. For this experiment we generate random orthogonal matrix $\mathbf{V}$ and random symmetric positive definite matrix $\tilde{\mathbf{I}}$, which satisfies $\tilde{\mathbf{I}}_{ii} = 1$, $\left|\tilde{\mathbf{I}}_{ij}\right| \leq \varepsilon$ for $i \neq j$. Columns of matrix $\mathbf{A}^{-1/2}\mathbf{V}\tilde{\mathbf{I}}^{1/2}$ were taken as approximate $\mathbf{A}$-conjugate vectors.

Figure 6: Expected precision $\mathbb{E}\left[\frac{||x_t - x_*||_{\mathbf{A}}^2}{||x_0 - x_*||_{\mathbf{A}}^2}\right]$ vs. the number of iterations of iSconD with different choices of parameter $\varepsilon$.

## E.4 Approximate solution without iterative methods

Note that the problem (1) is equivalent to the following problem of finding $x$ such that

$$\mathbf{A}x = b. \tag{32}$$

Let $\mathbf{S} = [v_1 \ldots v_n]$ be a set of $\mathbf{A}$-conjugate vectors, i.e., $\mathbf{S}^\top \mathbf{A} \mathbf{S} = \mathbf{I}$. We can now find the solution to the linear system (32). Since $\mathbf{S}^\top b = \mathbf{S}^\top \mathbf{A} x = \mathbf{S}^\top \mathbf{A} \mathbf{S} \mathbf{S}^{-1} x = \mathbf{S}^{-1} x$, we conclude that

$$x = \mathbf{S}\mathbf{S}^\top b. \tag{33}$$

We will now show that unlike in the exact case, using formula (33) with $\varepsilon$-approximate $\mathbf{A}$-conjugate vectors does *not* lead to a precise solution of our problem.

**Lemma 16.** *Let $\mathbf{S} = [v_1 \ldots v_n]$ be an $\varepsilon$-$\mathbf{A}$-orthonormal system. Let $x_* = \mathbf{A}^{-1} b$ be the solution of the linear system* (32). *Let $\hat{x}$ be an estimate of the solution, calculated with formula* (33) *using $\varepsilon$-approximate $\mathbf{A}$-conjugate vectors: $\hat{x} = \mathbf{S}\mathbf{S}^\top b$. If $\varepsilon < 1/(n-1)$, then*

$$\|\hat{x} - x_*\|_{\mathbf{A}} \leq \varepsilon(n-1) \frac{1 + \varepsilon(n-1)}{1 - \varepsilon(n-1)} \|x_*\|_{\mathbf{A}}$$

*Proof.* Note that $\mathbf{A}^{1/2}\hat{x} = \mathbf{A}^{1/2}\mathbf{S}\mathbf{S}^\top \mathbf{A}^{1/2}\mathbf{A}^{1/2}x_* = \hat{\mathbf{I}}\mathbf{A}^{1/2}x_*$, where $\hat{\mathbf{I}} = \mathbf{A}^{1/2}\mathbf{S}\mathbf{S}^\top \mathbf{A}^{1/2}$. From Lemma 14 we now get that

$$\left| \lambda_i(\hat{\mathbf{I}} - \mathbf{I}) \right| \leq \varepsilon(n-1) \frac{1 + \varepsilon(n-1)}{1 - \varepsilon(n-1)},$$

and hence

$$\left\| \hat{\mathbf{I}} - \mathbf{I} \right\|_2 \leq \varepsilon(n-1) \frac{1 + \varepsilon(n-1)}{1 - \varepsilon(n-1)}.$$

Therefore,

$$\|\hat{x} - x_*\|_{\mathbf{A}} = \left\| \mathbf{A}^{1/2}(\hat{x} - x_*) \right\|_2 = \left\| (\hat{\mathbf{I}} - \mathbf{I})\mathbf{A}^{1/2}x_* \right\|_2 \leq \left\| \hat{\mathbf{I}} - \mathbf{I} \right\|_2 \left\| \mathbf{A}^{1/2}x_* \right\|_2 \leq$$

$$\leq \varepsilon(n-1) \frac{1 + \varepsilon(n-1)}{1 - \varepsilon(n-1)} \|x_*\|_{\mathbf{A}}.$$

$\square$

If we choose $\varepsilon = \frac{1}{3(n-1)}$, like we did in Theorem 15, we get the following precision:

$$\|\hat{x} - x_*\|_{\mathbf{A}} \leq \frac{2}{3} \|x_*\|_{\mathbf{A}},$$

which is rather poor. However if we use Algorithm 7, we can get approximate solution with any precision (after enough iterations).

# F  Inexact SSD: a method that is not a special case of stochastic descent

In Section 2.1 we defined Stochastic Spectral Descent (Algorithm 2). We now design a new method which will "try" to use the same iterations, but with *inexact* eigenvectors of $\mathbf{A}$. We call $w$ an inexact eigenvector of $\mathbf{A}$ if

$$\mathbf{A}w = \lambda w + \varepsilon \tag{34}$$

for some $\varepsilon$ and $\lambda > 0$ (inexact eigenvalue). Clearly, *any* vector can be written in the form (34). This idea leads to Algorithm 8.

---

**Algorithm 8** Inexact Stochastic Spectral Descent (iSSD)

---

**Initialize:** $x_0 \in \mathbb{R}^n$; $(w_1, \lambda_1), \ldots (w_n, \lambda_n)$: inexact eigenvectors and eigenvalues of $\mathbf{A}$
**for** $t = 0, 1, 2, \ldots$ **do**
    Choose $i \in [n]$ uniformly at random
    Set $x_{t+1} = x_t - \left( w_i^\top x_t - \frac{w_i^\top b}{\lambda_i} \right) w_i$
**end for**

---

Note that the above method is *not equivalent* to applying stochastic descent $\mathcal{D}$ being the uniform distribution over the inexact eigenvectors. This is because in arriving at SSD, we have used some

properties of the eigenvectors and eigenvalues to simplify the calculation of the stepsize. The same simplifications do *not* apply for inexact eigenvectors. Nevertheless, we can formally run SSD, as presented in Algorithm 2, and replace the exact eigenvectors and eigenvalues by inexact versions thereof, thus capitalizing on the fast computation of stepsize which positively affects the cost of one iteration of the method. This leads to Algorithm 8.

Hence, in order to analyze the above method, we need to develop a completely new approach. We will show that Algorithm 8 converges only to a neighbourhood of the optimal solution.

## F.1   Lemmas

Further we will use the following notation: $\mathbf{S} = [w_1 \dots w_n]$ – inexact eigenvectors matrix, $\Lambda = \mathrm{Diag}\left(\lambda_1 \dots \lambda_n\right)$ – inexact eigenvalues matrix, $\mathbf{E} = [\varepsilon_1 \dots \varepsilon_n]$ – error matrix, $\tilde{\mathbf{A}} = \mathbf{S}\Lambda\mathbf{S}^\top$ – estimation of matrix $\mathbf{A}$. We also assume, that inexact eigenvectors are $\varepsilon$-approximate orthonormal for $\varepsilon < \frac{1}{n-1}$, i.e. $w_i^\top w_i = 1$, $\left|w_i^\top w_j\right| \leq \varepsilon$ for $i \neq j$.

The following lemma gives an answer to the question: how precise is $\tilde{\mathbf{A}}$ as an estimate of matrix $\mathbf{A}$?

**Lemma 17.** $\tilde{\mathbf{A}} = \hat{\mathbf{I}}\mathbf{A} - \mathbf{S}\mathbf{E}^\top$, *where matrix* $\hat{\mathbf{I}} = \mathbf{S}\mathbf{S}^\top$ *satisfies*

$$\left\|\hat{\mathbf{I}} - \mathbf{I}\right\|_2 \leq \varepsilon(n-1)\frac{1 + \varepsilon(n-1)}{1 - \varepsilon(n-1)}. \tag{35}$$

*Proof.* Indeed, the definition of inexact eigenvectors can be written in matrix form as $\mathbf{A}\mathbf{S} = \mathbf{S}\Lambda + \mathbf{E}$, from which follows that $\hat{\mathbf{I}}\mathbf{A} = \mathbf{S}\mathbf{S}^\top\mathbf{A} = \mathbf{S}\Lambda\mathbf{S}^\top + \mathbf{S}\mathbf{E}^\top$. Equality (35) follows immediately from Lemma 14. $\square$

The next lemma gives a general recursion capturing one step of iSSD, shedding light on the convergence of the method.

**Lemma 18.** *Sequence of $\{x_t\}$ generated by inexact SSD satisfies equality*

$$
\begin{aligned}
\mathbb{E}\left\|x_{t+1} - x_*\right\|_{\mathbf{A}}^2 \;=\; & \left(1 - \frac{1}{n}\right)\mathbb{E}\left\|x_t - x_*\right\|_{\mathbf{A}}^2 + \frac{1}{n}\mathbb{E}\left[(x_t - x_*)^\top\Gamma(x_t - x_*)\right] \\
& + \;\frac{1}{n}\left(\mathbb{E}\left\|x_t\right\|_{\mathbf{E}\Lambda^{-1}\mathbf{E}^\top}^2 + x_*^\top\mathbf{E}\Lambda^{-2}\mathbf{C}\mathbf{E}^\top x_*\right) \\
& - \;\frac{2}{n}\mathbb{E}\left[(x_t - x_*)^\top\mathbf{S}\mathbf{C}\Lambda^{-1}\mathbf{E}^\top x_*\right],
\end{aligned}
$$

*where* $\Gamma = (\mathbf{I} - \hat{\mathbf{I}})\mathbf{A} - \mathbf{S}\mathbf{E}^\top - \mathbf{E}\Lambda^{-1}\mathbf{E}^\top + \mathbf{S}\mathbf{C}\mathbf{S}^\top$ *and*

$$\mathbf{C} = \mathrm{Diag}\left(w_1^\top\varepsilon_1 \dots w_n^\top\varepsilon_n\right). \tag{36}$$

*Proof.*

$$\|x_{t+1} - x_*\|_{\mathbf{A}}^2 = \left\| x_t - x_* - \omega w_t w_t^\top (x_t - x_*) + \omega \frac{\varepsilon_t^\top x_*}{\lambda_t} w_t \right\|_{\mathbf{A}}^2$$

$$= \|x_t - x_*\|_{\mathbf{A}}^2 + \omega^2 w_t^\top \mathbf{A} w_t \left( w_t^\top (x_t - x_*) - \frac{\varepsilon_t^\top x_*}{\lambda_t} \right)^2$$

$$+ 2\omega (x_t - x_*)^\top \mathbf{A} w_t \left( \frac{\varepsilon_t^\top x_*}{\lambda_t} - w_t^\top (x_t - x_*) \right)$$

$$= \|x_t - x_*\|_{\mathbf{A}}^2 + \omega^2 (\lambda_t + w_t^\top \varepsilon_t) \left( w_t^\top (x_t - x_*) - \frac{\varepsilon_t^\top x_*}{\lambda_t} \right)^2$$

$$+ 2\omega (x_t - x_*)^\top (\lambda_t w_t + \varepsilon_t) \left( \frac{\varepsilon_t^\top x_*}{\lambda_t} - w_t^\top (x_t - x_*) \right)$$

$$= \|x_t - x_*\|_{\mathbf{A}}^2 - \omega(2-\omega)(x_t - x_*)^\top \lambda_t w_t w_t^\top (x_t - x_*) + \omega^2 \frac{x_*^\top \varepsilon_t \varepsilon_t^\top x_*}{\lambda_t}$$

$$+ 2\omega \frac{(x_t - x_*)^\top \varepsilon_t \varepsilon_t^\top x_*}{\lambda_t} + \omega^2 w_t^\top \varepsilon_t \left( w_t^\top (x_t - x_*) - \frac{\varepsilon_t^\top x_*}{\lambda_t} \right)^2$$

$$+ 2(\omega - \omega^2)(x_t - x_*)^\top w_t \varepsilon_t^\top x_* - 2\omega (x_t - x_*)^\top w_t \varepsilon_t^\top (x_t - x_*)$$

$$= \|x_t - x_*\|_{\mathbf{A}}^2 - \omega(2-\omega)(x_t - x_*)^\top \lambda_t w_t w_t^\top (x_t - x_*)$$

$$+ \quad \|x_*(\omega - 1) + x_t\|_{\frac{\varepsilon_t \varepsilon_t^\top}{\lambda_t}} - \|x_t - x_*\|_{\frac{\varepsilon_t \varepsilon_t^\top}{\lambda_t}}$$

$$+ \quad 2\omega (x_t - x_*)^\top w_t \varepsilon_t^\top (x_*(2-\omega) - x_t)$$

$$+ \quad \omega^2 w_t^\top \varepsilon_t \left( w_t^\top (x_t - x_*) - \frac{\varepsilon_t^\top x_*}{\lambda_t} \right)^2 .$$

Now we can take conditional expectation $\mathbb{E}[\,\cdot\mid x_t]$.

$$\mathbb{E}[\|x_{t+1} - x_*\|_{\mathbf{A}}^2 \mid x_t] = \|x_t - x_*\|_{\mathbf{A}}^2 - \frac{\omega(2-\omega)}{n} \|x_t - x_*\|_{\tilde{\mathbf{A}}}^2 + \frac{1}{n} \|x_*(\omega - 1) + x_t\|_{\Sigma}^2$$

$$- \quad \frac{1}{n} \|x_t - x_*\|_{\Sigma}^2 - \frac{2\omega}{n} (x_t - x_*)^\top \mathbf{S} \mathbf{E}^\top (x_t - (2-\omega)x_*)$$

$$+ \quad \frac{\omega^2}{n} \sum_{i=1}^{n} w_i^\top \varepsilon_i \left( w_i^\top (x_t - x_*) - \frac{\varepsilon_i^\top x_*}{\lambda_i} \right)^2 ,$$

where $\Sigma = \mathbf{E}\Lambda^{-1}\mathbf{E}^\top$.

Now we set $\omega = 1$ and use Lemma 17.

$$\mathbb{E}[\|x_{t+1} - x_*\|_{\mathbf{A}}^2 \mid x_t] = \|x_t - x_*\|_{\mathbf{A}}^2 - \frac{1}{n}\|x_t - x_*\|_{\hat{\mathbf{A}}}^2 + \frac{1}{n}\|x_t\|_{\Sigma}^2 - \frac{1}{n}\|x_t - x_*\|_{\Sigma}^2 - $$

$$-\frac{2}{n}(x_t - x_*)^\top \mathbf{SE}^\top(x_t - x_*) + \frac{1}{n}\sum_{i=1}^{n} w_i^\top \varepsilon_i \left(w_i^\top(x_t - x_*) - \frac{\varepsilon_i^\top x_*}{\lambda_i}\right)^2 = $$

$$= \|x_t - x_*\|_{\mathbf{A}}^2\left(1 - \frac{1}{n}\right) + \frac{1}{n}(x_t - x_*)^\top\left((\mathbf{I} - \hat{\mathbf{I}})\mathbf{A} + \mathbf{SE}^\top - 2\mathbf{SE}^\top\right)(x_t - x_*)$$

$$+\frac{1}{n}\|x_t\|_{\Sigma}^2 - \frac{1}{n}\|x_t - x_*\|_{\Sigma}^2 + \frac{1}{n}\sum_{i=1}^{n} w_i^\top \varepsilon_i \left(w_i^\top(x_t - x_*) - \frac{\varepsilon_i^\top x_*}{\lambda_i}\right)^2 = $$

$$= \left(1 - \frac{1}{n}\right)\|x_t - x_*\|_{\mathbf{A}}^2 + \frac{1}{n}(x_t - x_*)^\top\left((\mathbf{I} - \hat{\mathbf{I}})\mathbf{A} - \mathbf{SE}^\top - \Sigma\right)(x_t - x_*) + \frac{1}{n}\|x_t\|_{\Sigma}^2 + $$

$$+\frac{1}{n}\sum_{i=1}^{n} w_i^\top \varepsilon_i \left(w_i^\top(x_t - x_*) - \frac{\varepsilon_i^\top x_*}{\lambda_i}\right)^2 = $$

$$= \left(1 - \frac{1}{n}\right)\|x_t - x_*\|_{\mathbf{A}}^2 + \frac{1}{n}(x_t - x_*)^\top\left((\mathbf{I} - \hat{\mathbf{I}})\mathbf{A} - \mathbf{SE}^\top - \Sigma\right)(x_t - x_*) + \frac{1}{n}\|x_t\|_{\Sigma}^2 + $$

$$+\frac{1}{n}\|x_t - x_*\|_{\mathbf{SCS}^\top}^2 + \frac{1}{n}x_*^\top\left(\mathbf{E}\Lambda^{-2}\mathbf{CE}^\top\right)x_* - \frac{2}{n}(x_t - x_*)^\top\mathbf{SC}\Lambda^{-1}\mathbf{E}^\top x_*,$$

where $\mathbf{C} = \text{Diag}\left(w_1^\top \varepsilon_1 \ldots w_n^\top \varepsilon_n\right)$. We get

$$\mathbb{E}[\|x_{t+1} - x_*\|_{\mathbf{A}}^2 \mid x_t] = \left(1 - \frac{1}{n}\right)\|x_t - x_*\|_{\mathbf{A}}^2 + \frac{1}{n}(x_t - x_*)^\top\Gamma(x_t - x_*)$$

$$+\frac{1}{n}\left(\|x_t\|_{\mathbf{E}\Lambda^{-1}\mathbf{E}^\top}^2 + x_*^\top\mathbf{E}\Lambda^{-2}\mathbf{CE}^\top x_* - 2(x_t - x_*)^\top\mathbf{SC}\Lambda^{-1}\mathbf{E}^\top x_*\right),$$

where $\Gamma = (\mathbf{I} - \hat{\mathbf{I}})\mathbf{A} - \mathbf{SE}^\top - \mathbf{E}\Lambda^{-1}\mathbf{E}^\top + \mathbf{SCS}^\top$. $\qquad\square$

The following lemma describes which inexact eigenvalues are optimal for a fixed set of inexact eigenvectors.

**Lemma 19.** *Let $w_i$ be fixed. Then the choice*

$$\lambda_i = w_i^\top \mathbf{A} w_i \tag{37}$$

*minimizes $\|\varepsilon_i\|_2$ in $\lambda$, where $\epsilon_i := \|\mathbf{A} w_i - \lambda w_i\|_2$. Moreover, for this choice of $\lambda_i$ we get $w_i^\top \varepsilon_i = 0$.*

*Proof.* Minimizing $\|\mathbf{A} w_i - \lambda w_i\|_2^2$ in $\lambda$ gives (37). For this choice of $\lambda_i$ we get $w_i^\top \varepsilon_i = w_i^\top \mathbf{A} w_i - \lambda_i w_i^\top w_i = w_i^\top \mathbf{A} w_i - w_i^\top \mathbf{A} w_i = 0$. $\qquad\square$

### F.2 Convergence

Choosing eigenvalues as defined in (37), and in view of (36), we see that $\mathbf{C} = 0$. From this and Lemma 18 we get

$$\mathbb{E}\|x_{t+1} - x_*\|_{\mathbf{A}}^2 = \left(1 - \frac{1}{n}\right)\mathbb{E}\|x_t - x_*\|_{\mathbf{A}}^2 + \frac{1}{n}\mathbb{E}\left[(x_t - x_*)^\top\Gamma(x_t - x_*)\right] + \frac{1}{n}\mathbb{E}\|x_t\|_{\mathbf{E}\Lambda^{-1}\mathbf{E}^\top}^2,$$

where $\Gamma = (\mathbf{I} - \hat{\mathbf{I}})\mathbf{A} - \mathbf{SE}^\top - \mathbf{E}\Lambda^{-1}\mathbf{E}^\top$. From the Cauchy–Schwarz inequality we get

$$\frac{1}{n}\mathbb{E}\|x_t\|_{\mathbf{E}\Lambda^{-1}\mathbf{E}^\top}^2 = \frac{1}{n}\mathbb{E}\|x_t - x_* + x_*\|_{\mathbf{E}\Lambda^{-1}\mathbf{E}^\top}^2 \leq $$

$$\leq \frac{2}{n}\mathbb{E}\|x_t - x_*\|_{\mathbf{E}\Lambda^{-1}\mathbf{E}^\top}^2 + \frac{2}{n}\mathbb{E}\|x_*\|_{\mathbf{E}\Lambda^{-1}\mathbf{E}^\top}^2,$$

which leads to

$$\mathbb{E}\|x_{t+1} - x_*\|_{\mathbf{A}}^2 \leq \left(1 - \frac{1}{n}\right)\mathbb{E}\|x_t - x_*\|_{\mathbf{A}}^2 + \frac{1}{n}\mathbb{E}\left[(x_t - x_*)^\top\mathbf{Q}(x_t - x_*)\right] + \frac{2}{n}\|x_*\|_{\mathbf{E}\Lambda^{-1}\mathbf{E}^\top}^2, \tag{40}$$

where $\mathbf{Q} = (\mathbf{I} - \hat{\mathbf{I}})\mathbf{A} - \mathbf{SE}^\top + \mathbf{E}\Lambda^{-1}\mathbf{E}^\top$. Inequality (40) implies that

$$\mathbb{E} \left\| x_{t+1} - x_* \right\|_{\mathbf{A}}^2 \leq \mathbb{E} \left\| x_t - x_* \right\|_{\mathbf{A}}^2 + \frac{q-1}{n} \mathbb{E} \left\| x_t - x_* \right\|_{\mathbf{A}}^2 + \frac{r_0}{n}, \qquad (41)$$

where $q = \max \frac{z^\top \mathbf{Q} z}{z^\top \mathbf{A} z}$, $r_0 = 2 \left\| x_* \right\|_{\mathbf{E}\Lambda^{-1}\mathbf{E}^\top}^2$.

If the errors $\varepsilon_1, \dots, \varepsilon_n$ and $\varepsilon$ are small enough, we can make $q$ and $r_0$ arbitrarily small for fixed $x_*$. From (41) we can see that $\mathbb{E} \left\| x_{t+1} - x_* \right\|_{\mathbf{A}}^2$ is going to decrease as long as

$$\mathbb{E} \left\| x_t - x_* \right\|_{\mathbf{A}}^2 \geq \frac{r_0}{1-q}. \qquad (42)$$

Hence, for small enough $\varepsilon_1, \dots, \varepsilon_n$ and parameter $\varepsilon$, iSSD will converge to a neighborhood of the optimal solution, with limited precision (42).