[Reviews · NeurIPS 2018]

Reviewer 1



This paper focuses on quadratic minimization and proposes a novel stochastic gradient descent algorithm by interpolating randomized coordinate descent (RCD) and stochastic spectral descent (SSD). Those two algorithms were studied in prior but they have drawbacks to apply for practical usage. Hence, authors combine them by random coin-tossing (i.e., with probability \alpha run RCD and with probability 1-\alpha run SSD) and provide the optimal parameters to achieve the best convergence rate. In addition, they study the rate of their algorithm for some special cases, e.g., a matrix in objective has clustered or exponentially decaying eigenvalues. In experiments, they report practical behavior of their algorithm under synthetic settings. The proposed algorithm takes advantages of two prior works, i.e., efficient gradient update from RCD and fast convergence from SSD. And it is interesting that the rate of the algorithm is between that of RCD and SSD, but not surprising. Moreover, authors study some extreme cases for RCD that the rate is arbitrary large and it is questionable that the proposed algorithm have the cases. It would be better to mention some extreme cases for SSCD. And authors benchmark their algorithm under only synthetic dataset. It is desired to test the algorithm under real-world dataset and report its performance. Furthermore, there have been studied many algorithms for quadratic minimization, the proposed algorithm should compete with other works. Overall, the main contribution of this paper is interesting with an intuitive interpolating approach and some case studies. However, the limited experiments make this work weaken. I consider this works as marginally above the acceptance. ***** I have read the author feedback. I appreciate authors for providing the details about my questions. *****

Reviewer 2



The authors develop a new way of accelerating the coordinate descent method by using the spectral decomposition of the gradient matrix and conjugate direction. I personally think that this approach is interesting and has potentials in solving large scaled problems. Apart from a linear rate, they show that this rate is independent of the condition number, which I find it very interesting. However, from what I am seeing, the main results of this paper could only be applied for the smooth minimization problem. It would be nice if the authors could confirm about some of their extensions for non-smooth optimization.

Reviewer 3



The goal of this paper is to introduce a new acceleration strategy for randomized coordinate descent (RCD) methods. This is achieved by taking the standard coordinate directions and `enriching' them with a few spectral or conjugate directions. The authors specifically consider the problem stated in (1), which is the minimization of a quadratic function involving a symmetric and positive definite matrix (so the optimal solution is unique). RCD applied to (1) is described in Algorithm 3, Stochastic Spectral Descent (SSD) is described in Algorithm 2, and complexity results for both algorithms are given. The main point that the authors stress is that for RCD, the complexity result depends upon the problem conditioning, whereas for SSD the complexity result is independent of conditioning and depends only on the problem dimension. The SSD algorithm, is theoretically interesting, but is impractical. With this in mind, the authors describe a practical algorithm called Spectral Stochastic Coordinate Descent (SSCD) that uses just a few spectral directions in addition to the standard coordinate directions, and provides a complexity result for SSCD, stating that the algorithm converges linearly. Numerical results are also given to show the practical performance of the algorithm. The paper is clear, well organized, and while it only focuses on a specific problem (quadratic minimization) the treatment is thorough. (And the authors explain that they have restricted their attention to this case because they already have enough to say on this topic, but that extensions to other cases have been considered and that work is ongoing.) I believe that this paper will be of interest to the NIPS community.